# Patterns and determinants of the global herbivorous mycobiome

Casey H. Meili [1], Adrienne L. Jones[1], Alex X. Arreola[1], Jeffrey Habel[1], Carrie J. Pratt [1], Radwa A. Hanafy[1], Yan Wang [2], Aymen S. Yassin [3], Moustafa A. TagElDein[3], Christina D. Moon[4], Peter H. Janssen [4], Mitesh Shrestha[5], Prajwal Rajbhandari [5], Magdalena Nagler [6], Julia M. Vinzelj [6], Sabine M. Podmirseg[6], Jason E. Stajich [7], Arthur L. Goetsch[8], Jerry Hayes[8], Diana Young[9], Katerina Fliegerova [10], Diego Javier Grilli[11], Roman Vodička[12], Giuseppe Moniello[13], Silvana Mattiello [14], Mona T. Kashef[3], Yosra I. Nagy[3], Joan A. Edwards [15], Sumit Singh Dagar[16], Andrew P. Foote [17], Noha H. Youssef [1] ✉ & Mostafa S. Elshahed [1] ✉

Despite their role in host nutrition, the anaerobic gut fungal (AGF) component of the herbivorous gut microbiome remains poorly characterized. Here, to examine global patterns and determinants of AGF diversity, we generate and analyze an amplicon dataset from 661 fecal samples from 34 mammalian species, 9 families, and 6 continents. We identify 56 novel genera, greatly expanding AGF diversity beyond current estimates (31 genera and candidate genera). Community structure analysis indicates that host phylogenetic affiliation, not domestication status and biogeography, shapes the community rather than. Fungal-host associations are stronger and more specific in hindgut fermenters than in foregut fermenters. Transcriptomics-enabled phylogenomic and molecular clock analyses of 52 strains from 14 genera indicate that most genera with preferences for hindgut hosts evolved earlier (44-58 Mya) than those with preferences for foregut hosts (22-32 Mya). Our results greatly expand the documented scope of AGF diversity and provide an ecologically and evolutionary-grounded model to explain the observed patterns of AGF diversity in extant animal hosts.

Plant biomass represents the most abundant[1], yet least readily digestible[2] nutritional source on Earth. The rise of herbivory in tetrapods was associated with multiple evolutionary innovations to maximize plant biomass degradation efficiency[3,4]. This allowed for longer food retention times as well as the acquisition and retention of an endosymbiotic anaerobic microbial community, both of which enhance the breakdown of ingested plant material and increase feed energy supply to the host in the form of fermentation products[5,6]. Extant families of mammalian herbivores are characterized by the enlargement of portions of their gut, where the majority of fermentation of plant material occurs. Animals are classified based on their fermentation sites into hindgut fermenters (e.g., members of the families Equidae and Elephantidae, where an enlarged colon, caecum, or rectum constitutes the fermentation chamber and harbors the fermentative community), or foregut fermenters (where a pre-gastric fermentation chamber is enlarged). Foregut fermenters are in turn classified according to the anatomy of their pre-gastric fermentation chamber into pseudoruminant foregut fermenters (e.g. members of Camelidae with enlarged diverticula or fermentative sacs), and foregut ruminants (e.g. members of Bovidae and Cervidae with a more complex four-chambered stomach, with the rumen being the largest chamber and where plant material degradation and fermentation

occurs) (Fig. 1a)[5,7,8]. Within the highly diverse microbial consortia residing in the expanded herbivorous alimentary tract, the anaerobic gut fungi (AGF, phylum Neocallimastigomycota) were the last to be recognized[9–11] and remain the most enigmatic. In spite of their critical role in initiating plant biomass colonization[12,13], their wide array of highly efficient lignocellulolytic enzymes[14–23], and their biotechnological potential[24–26], AGF diversity and distribution patterns remain, to-date, very poorly characterized[27]. Culture-independent efforts targeting AGF have long been hampered by the documented shortcomings of the universal fungal ITS1 barcoding marker for accurately characterizing AGF diversity[27,28] and, until recently, by the lack of clear thresholds and procedures for genus and species OTUs delineation[29]. Recent efforts have indeed started to yield interesting insights on AGF diversity. However, the number of high-throughput diversity surveys targeting AGF conducted so far remains limited, compared to surveys of their bacterial and archaeal counterparts (Table S1). Further, most prior studies were limited in scope and/or breadth, usually analyzing a limited number of samples from a single or few mostly domesticated hosts residing in a single location. Given the large number of extant putative mammalian hosts (e.g. the family Bovidae comprises 8 subfamilies, more than 50 genera, and 143 extant species[30]), as well as the immense number of herbivorous mammals on Earth (a conservative estimate of 75.3 million wild, and 3.5 billion domesticated ruminants, including ≈1.4 billion cattle, 1.1 billion sheep, 0.85 billion goats, ~60 million horses[31], and ~50 million donkeys[32]), it is clear that the global AGF diversity remains severely under-sampled.

Beyond documenting diversity and identifying novel lineages, the current patchy and incomplete view of AGF diversity precludes any systematic analysis of the patterns (distribution, relative abundance, and AGF taxa distribution preferences) and determinants (role of and interplay between various factors in structuring communities) of the global herbivorous mycobiome. Assembly and structuring of microbial communities could be governed by deterministic (niche theory-based) or stochastic (null theory-based) processes[33]. The co-occurrence and dynamic interplay between deterministic and stochastic processes is increasingly being recognized[33–35]. Stochastic processes generate changes in community diversity that would not be distinguishable from those changes produced by random chance and include dispersal (movement of organisms from one location to another with subsequent successful colonization in the new location), and drift (defined as random changes in relative abundances of species or individuals due to stochastic factors such as birth, death, or multiplication). Possible deterministic processes governing AGF community assembly include animal host identity (family, species), and gut-type (foregut ruminant, fermenting pseudoruminant, and hindgut fermenters). Beyond host-associated factors, AGF communities could also be impacted by the host domestication status (i.e., whether reared in a domesticated setting and hence predominantly grazers on grasses, or are wild and hence predominantly browsers for fruits, shoots, shrubs, forbs, and tree leaves diets[31]), as well as biogeography, age, sex, or local feed chemistry.

To assess global patterns and determinants of AGF diversity, a consortium of scientists from 17 institutions have sampled fecal material from domesticated and non-domesticated animals from 6 continents covering 9 mammalian families, and 3 gut types. The dataset obtained was used to document the scope of AGF diversity on a global scale and to assess evolutionary and ecological drivers shaping AGF diversity and community structure using the large ribosomal subunit (LSU) as a phylogenetic marker[27]. Furthermore, to assess the evolutionary drivers underpinning the observed pattern of animal host-AGF phylosymbiosis, a parallel transcriptomics sequencing effort for 20 AGF strains from 13 genera was conducted and combined with previous efforts[36–41]. The expanded AGF transcriptomic dataset (52 strains from 14 genera) enabled phylogenomic and molecular timing analysis that correlated observed ecological patterns with fungal and

hosts evolutionary histories. Our results greatly expand the scope of documented AGF diversity, demonstrate the complexity of ecological processes shaping AGF communities, and demonstrate that host-specific evolutionary processes (e.g., evolution of host families, genera, and gut architecture) played a key role in driving a parallel process of AGF evolution and diversification.

## Results
### Sampling overview
A total of 661 samples belonging to 34 species and 9 families of foregut-fermenting ruminant (thereafter ruminant, $n = 468$), foregut-fermenting pseudoruminant (thereafter pseudoruminant, $n = 17$), and hindgut fermenters (n = 176) were examined (Fig. 1a, b, Supplementary Data 1). The dataset also provides a high level of replication for a variety of animals (229 cattle, 138 horses, 96 goats, 71 sheep, and 23 white-tail deer, among others) (Fig. 1b), locations (418 samples from USA, 74 from Egypt, 38 from Italy, 35 from New Zealand, 31 from Germany, and 25 from Nepal, among others) (Fig. 1a, Supplementary Data 1), and domestication status (564 domesticated, 97 undomesticated) (Fig. 1b, Supplementary Data 1), allowing for robust statistical analysis. As well, many samples belong to previously unsampled or rarely sampled animal families (e.g., Caviidae, Trichechidae), and species e.g., capybara ($n = 3$), mara ($n = 2$), manatee ($n = 2$), chamois ($n = 2$), markhor ($n = 1$), and takin ($n = 1$). In spite of the sparse numbers and low replicability for these animals, their inclusion provides an opportunity to examine the diversity and community structure patterns, as well as potential novelty of AGF in these hitherto unsampled hosts.

### Amplicon sequencing overview
A total of 8.73 million Illumina sequences of the hypervariable region 2 of the large ribosomal subunit (D2 LSU) were obtained. Rarefaction curve (Fig. S1) and coverage estimates (Supplementary Data 2) demonstrated that the majority of genus-level diversity (>90% based on Good's coverage) was captured in 97.7% of samples. The overall composition of the dataset showed a high genus-level phylogenetic diversity, with representatives of 19 out of the 20, and 10 out of the 11, currently described genera, and yet-uncultured candidate genera, respectively, identified (Fig. 1c, d, S2, Supplementary Data 2). Ubiquity (number of samples in which a taxon is identified) and relative abundance (percentage of sequences belonging to a specific taxon) of different genera were largely correlated ($R^2 = 0.71$, Fig. S3).

To confirm that these patterns were not a function of the primer pair, or sequencing technology (Illumina) employed, we assessed the reproducibility of the observed patterns by conducting a parallel sequencing effort on the first batch of available samples ($n = 61$) using a different set of primers targeting the entire D1/D2 LSU region (~700 bp D1/D2), and a different sequencing technology (SMRT PacBio). A highly similar community composition was observed when comparing datasets generated from the same sample using Illumina versus SMRT technologies, as evidenced by small Euclidean distances on CCA ordination plot between each pair of Illumina versus PacBio sequenced sample (Fig. S4b–d). Ordination-based community structure analysis indicated that the sequencing method employed had no significant effect on the AGF community structure (Canonical correspondence analysis ANOVA $p$-value = 0.305) (Fig. S4).

### Expanding Neocallimastigomycota diversity
Interestingly, 996,374 sequences (11.4% of the total) were not affiliated with any of the 20 currently recognized AGF genera or 11 candidate genera. Detailed phylogenetic analysis grouped these unaffiliated sequences into 56 novel genera, designated NY1-NY56 (Fig. 2a), hence expanding AGF genus-level diversity by a factor of 2.75. In general, relative abundance of sequences affiliated with novel genera was higher in ruminants (Wilcoxon test adjusted $p$-value $< 2 \times 10^{-16}$), as well

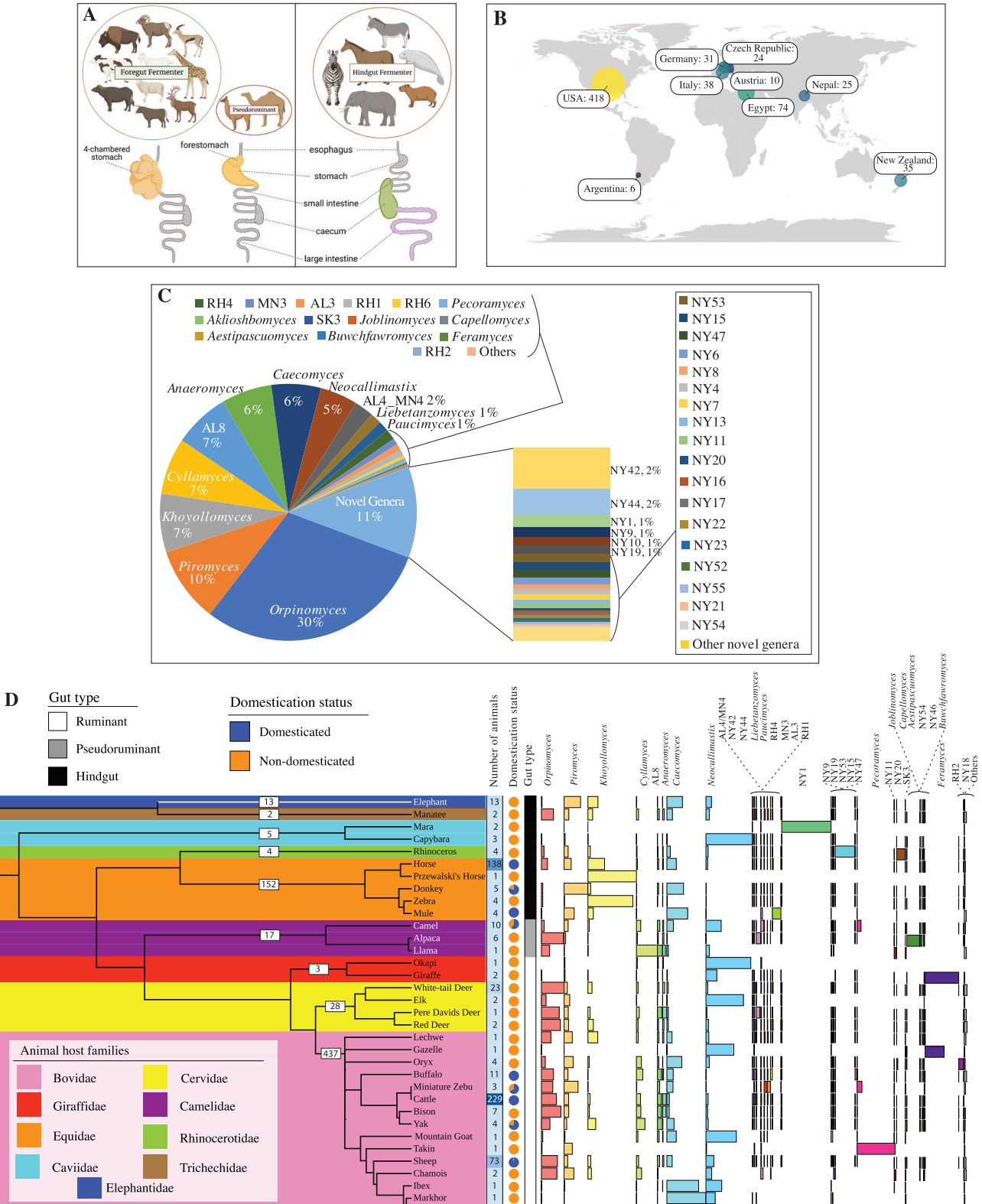

**Fig. 1 | Broad AGF diversity patterns in the herbivorous gut. A** Overview of the anatomy of the GIT tract of various types of herbivores. Colorized parts of the GIT indicate main location of plant fermentation for each gut type. **B** Map showing the geographical locations and the number of fecal samples analyzed in this study. **C** Pie chart showing the total percentage abundance of various AGF genera identified in the entire 8.73 million sequence dataset. Genera whose abundance never exceeded 1% in any of the samples are collectively designated as "others". **D** AGF community composition by animal species. The phylogenetic tree showing the relationship between animals was downloaded from timetree.org. The number of individuals sampled from each animal family is shown on the tree. The tracks to the right of the tree depict the number of individuals belonging to each animal species (shown as a heatmap with the actual number shown), domestication status, and the gut type. AGF community composition for each animal species is shown to the right as colored columns.

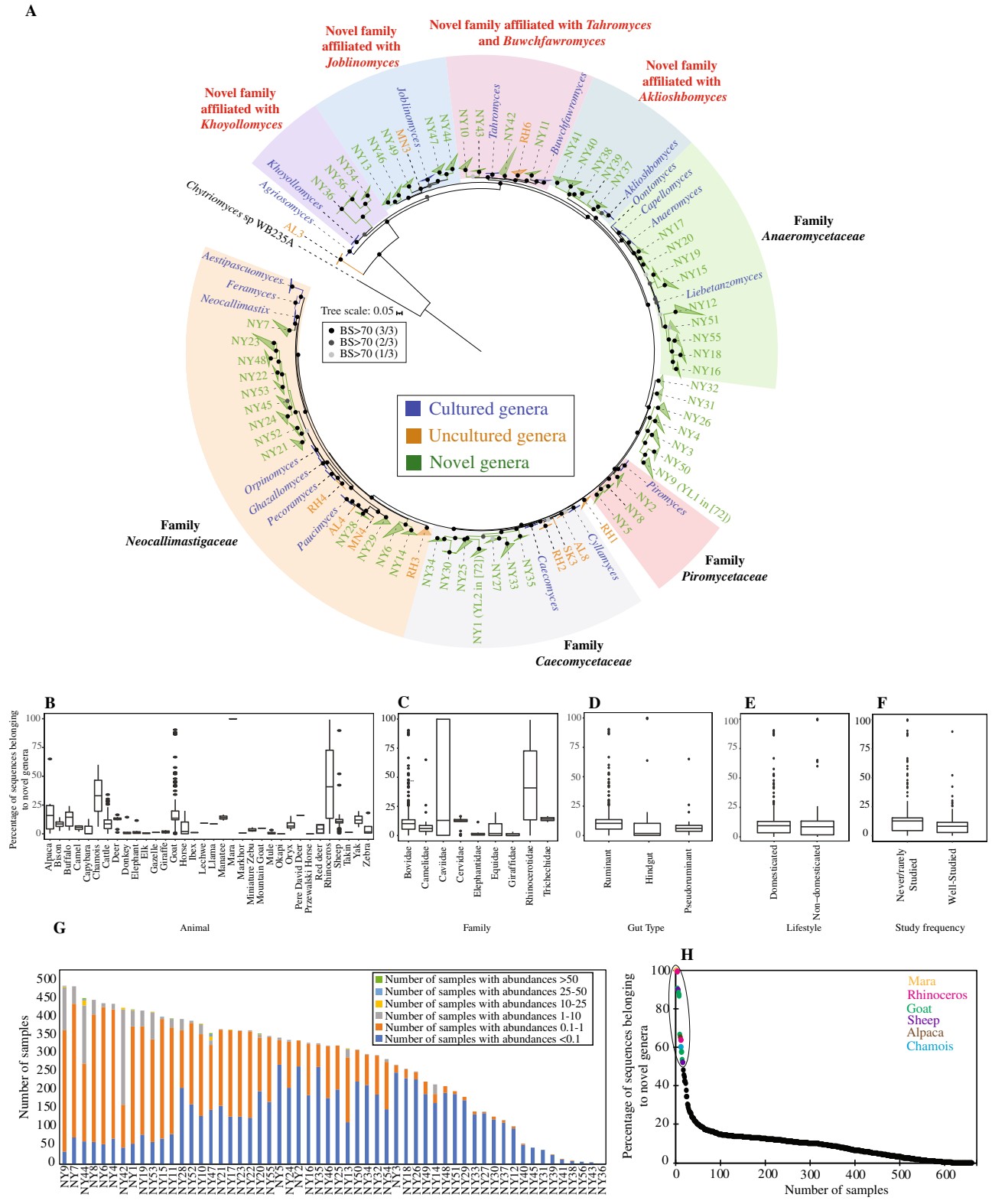

as in pseudoruminants (Wilcoxon test adjusted *p*-value = 0.02) compared to hindgut fermenters (Fig. 2b–d, Table S2). On the other hand, there was no significant difference in the relative abundance of novel genera based on domestication status (Wilcoxon test adjusted *p*-value = 0.69) (Fig. 2e, Table S2).

A closer look at the patterns of distribution of novel genera (Table S3) identified three important trends. First, the proportion of sequences belonging to novel genera in previously well-sampled

animals (cattle, sheep, goats, horses, and donkeys) was significantly smaller (Wilcoxon test adjusted *p*-value = 2.3 × 10⁻¹⁰) (Fig. 2f, h, Table S2) than in rarely sampled or previously unsampled hosts (e.g., buffalo, bison, deer, elephant, mara, capybara, manatee, among others), highlighting the importance of sampling hitherto unsampled or rarely sampled animals as a yet-unexplored reservoir for AGF diversity. Second, some novel genera were extremely rare and often identified solely in few sample replicates of a well-sampled animal (e.g., NY42,

**Fig. 2 | Expanding Neocallimastigomycota diversity. A** Maximum likelihood phylogenetic tree highlighting the position of novel AGF genera (NY1-NY56, green) identified in this study. The tree includes representatives from all previously reported cultured, and uncultured genera as references. Two of the 56 novel genera identified here correspond to two novel clades identified in a recent publication: NY1 corresponds to *Neocallimastigaceae* clade YL2, and NY9 corresponds to *Neocallimastigaceae* clade YL1 in ref. 98, and both names are given in the figure. Putative affiliations of novel identified genera with existing AGF families, affiliation with orphan genera, or position as completely novel families are highlighted. The three bootstrap support values (SH-aLRT, aBayes, and UFB) are shown as colored dots as follows: all three support values > 70%, black dot; 2/3 support values > 70%, dark grey; 1/3 support values > 70%, light grey. **B**–**F** Variation in the proportion of sequences affiliated with novel genera between different animal species, animal families, animal gut type, domestication status, and study frequency. Boxplots extend from the first to the third quartile and the median is shown as a thick line in the middle. The whiskers extending on both ends represent variability outside the quartiles and are calculated as follows: Minimum whisker=minimum quartile − 1.5 x inter-quartile range; Maximum whisker = maximum quartile + 1.5 x inter-quartile range. All points outside the box and whiskers are outliers. The number of data points used to calculate each box and whisker plot in (**B**, **C**) correspond to the number of samples belonging to each animal species, and animal family as defined in Fig. 1D. The number of data points used to calculate each box and whisker plot in (**D**–**F**) is shown on top of each plot. The results of Wilcoxon two-sided test of significance are shown in Table S2. **G**, **H** Distribution patterns of novel AGF genera identified in this study. **G** Number of samples with relative abundances of novel genera as shown in the figure legend to the right. **H** Percentage of sequences belonging to novel genera in each of the 661 samples. The 16 samples that harbored a community with >50% novel sequences are highlighted and color-coded by the animal species as shown in the key.

NY9, NY53, and NY17, in only 5, 2, 1, and 1 cattle samples, respectively) (Supplementary Data 2), highlighting the importance of replicate sampling for accurate assessment of hosts' novel pangenomic diversity. Finally, 5 of the 56 novel genera were never identified in >0.1% abundance in any sample (Fig. 2g, S2b), and 16 of the 56 never exceeded 1% (Fig. 2g, S2b), a pattern that highlights the value of deep sequencing to access perpetually rare members of the AGF community.

Phylogenetically, 32 of the novel genera identified clustered within the 4 recently proposed families in the Neocallimastigomycota[42], with 13, 7, 9, and 3 genera clustering with the families Neocallimastigaceae, Caecomycetaceae, Anaeromycetaceae, and Piromycetaceae, respectively. Another 17 novel genera formed 4 additional well-supported family-level clusters with orphan cultured genera, with 5, 4, 5, and 3 novel genera forming family-level clusters with the orphan genera *Joblinomyces, Buwchfawromyces-Tahromyces, Aklioshbomyces,* and *Khoyollomyces,* respectively. The remaining 7 novel genera were not affiliated with known cultured or uncultured genera and potentially formed novel family-level lineage(s) within the Neocallimastigomycota (Fig. 2a).

Confirmation of the occurrence of such an unexpectedly large number of novel AGF genera and simultaneous recovery of full-length sequence representatives (-700 bp covering the D1/D2 regions) was achieved by examining the SMRT-PacBio output generated from a subset (61 samples) of the total dataset, as described above. A total of 49 of the 56 novel genera were identified in the PacBio dataset (Supplementary Data 3). No additional new genera were found using this supplementary sequencing approach. Further, comparing SMRT- versus Illumina-generated tree topologies, revealed nearly identical topologies, phylogenetic distinction, and putative family-level assignments for all novel genera identified (Figs. S5, S6, Table S4).

### Stochastic and deterministic processes play an important role in shaping AGF community

Normalized stochasticity ratios (NST) were calculated based on two β-diversity indices (abundance-based Bray-Curtis index, and incidence-based Jaccard index. An NST value of >50% indicates a more stochastic assembly, while values < 50% indicate a more deterministic assembly[33]. NST values suggested that both stochastic and deterministic processes contribute to shaping AGF community assembly (Fig. 3a–h, Table S5). However, significant differences in the relative importance of these processes were observed across datasets regardless of the β-diversity index used. Specifically, hindgut fermenters and pseudoruminants exhibited significantly lower levels of stochasticity (hindgut: 58.03−56.5%; pseudoruminants: 65.4−79.3%) when compared to ruminants (81.7−86.4%) (Fig. 3a, e). This was also reflected at the animal family level (Fig. 3b, f), as well as at the animal species level (Fig. 3c, g). On the other hand, NST values were highly similar for domesticated versus non-domesticated animals (Fig. 3d, h). To further quantify the

contribution of specific deterministic (homogenous and heterogenous selection) and stochastic (homogenizing dispersal, dispersal limitation, and drift) processes in shaping the AGF community assembly, we used a two-step null-model-based quantitative framework that makes use of both taxonomic ($RC_{Bray}$) and phylogenetic (βNRI) β-diversity metrics[34,35]. Results (Fig. 3i) confirmed a lower overall level of stochasticity in hindgut fermenters, similar to the patterns observed using NST values. More importantly, the results indicate that homogenous selection (i.e., selection of specific taxa based on distinct differences between examined niches) represents the sole (99.8%) deterministic process shaping community assembly across all datasets (Fig. 3i). Within stochastic processes, drift played the most important role in shaping community assembly (83.4% of all stochastic processes), followed by homogenizing dispersal (16.6% of all stochastic processes), with a negligible contribution of dispersal limitation. As such, homogenous selection, drift, and homogenizing dispersal collectively represented the absolute (>99%), drivers of AGF community assembly, albeit with different relative importance of the three processes in datasets belonging to different animal species, family, gut type, or lifestyle (Fig. 3i).

### Community structure analysis reveals a strong pattern of fungal-host phylosymbiosis

Assessment of alpha diversity patterns indicated that gut type, animal family, and animal species, but not domestication status, significantly affected alpha diversity (Fig S7). Hindgut fermenters harbored a significantly less diverse community compared to ruminants. Within ruminants, no significant differences in alpha diversity levels were observed across various families (Cervidae and Bovidae) or species (deer, goat, cattle, and sheep) (Fig. S7).

Patterns of AGF community structure were assessed using ordination plots (PCoA, NMDS, and RDA) constructed using phylogenetic similarity-based (unweighted and weighted Unifrac) beta diversity indices (PCoA, and NMDS), or genera abundance data (RDA). The results demonstrated that host-associated factors (gut type, animal family, animal species) play a more important role in shaping the AGF community structure (Fig. 4, S8) when compared to domestication status, with samples broadly clustering by the gut type (Fig. 4c), and within this by animal family (Fig. 4b), and animal species (Fig. 4a). PERMANOVA results demonstrated that, regardless of the beta diversity measure, all factors significantly explained diversity (F statistic *p*-value = 0.001), with animal species explaining the most variance (14.7−21% depending on the index used), followed by animal family (6.5−7.2%), and animal gut type (5−5.4%). Host domestication status only explained 0.24−0.53% of variance and was not found to be significant with unweighted Unifrac (F statistic *p*-value = 0.143) (Fig. 4d).

Due to the inherent sensitivity of PERMANOVA to heterogeneity of variance among groups[43], we used three additional matrix

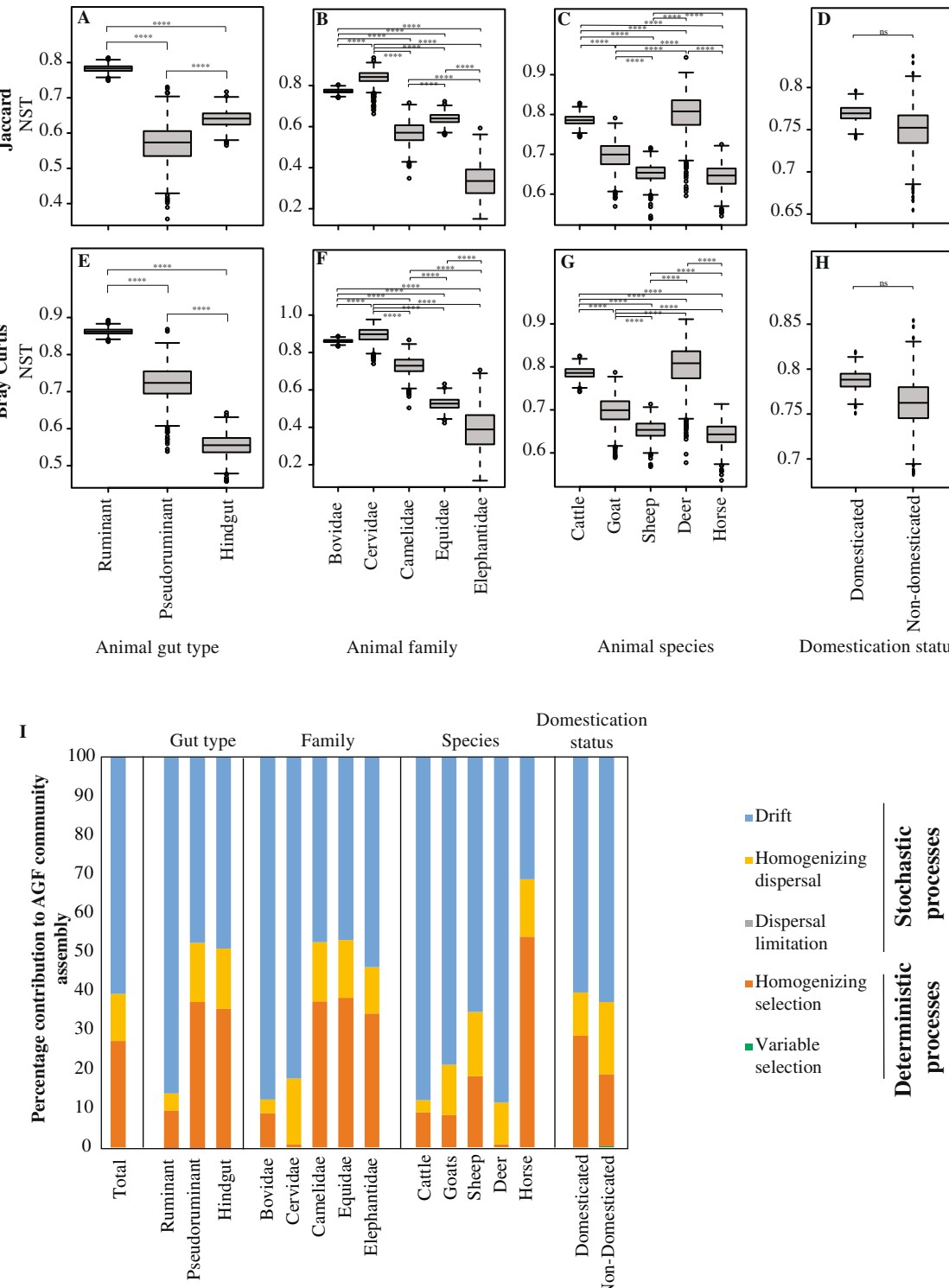

**Fig. 3 | Contribution of stochastic and deterministic processes to AGF community assembly. A–H** Levels of stochasticity in AGF community assembly were compared between different gut types (**A**, **E**), animal families (**B**, **F**; for families with more than 10 individuals), animal species (**C**, **G**; for animals with more than 20 individuals), and animal domestication status (**D**, **H**). Two normalized stochasticity ratios (NST) were calculated; the incidence-based Jaccard index (**A–D**), and the abundance-based Bray-Curtis index (**E–H**). Boxplots extend from the first to the third quartile and the median is shown as a thick line in the middle. The whiskers extending on both ends represent variability outside the quartiles and are calculated as follows: Minimum whisker=minimum quartile−1.5 x inter-quartile range; Maximum whisker=maximum quartile+1.5 x inter-quartile range. All points outside the box and whiskers are outliers. The box and whisker plots show the distribution of the bootstrapping results ($n = 1000$). ****: Wilcoxon two-sided $p$-value = $2 \times 10^{-16}$; ns not significant. **I** The percentages of the various deterministic and stochastic processes shaping AGF community assembly of the total dataset, and when sub-setting for different animal gut types, animal families, animal species, and animal lifestyles.

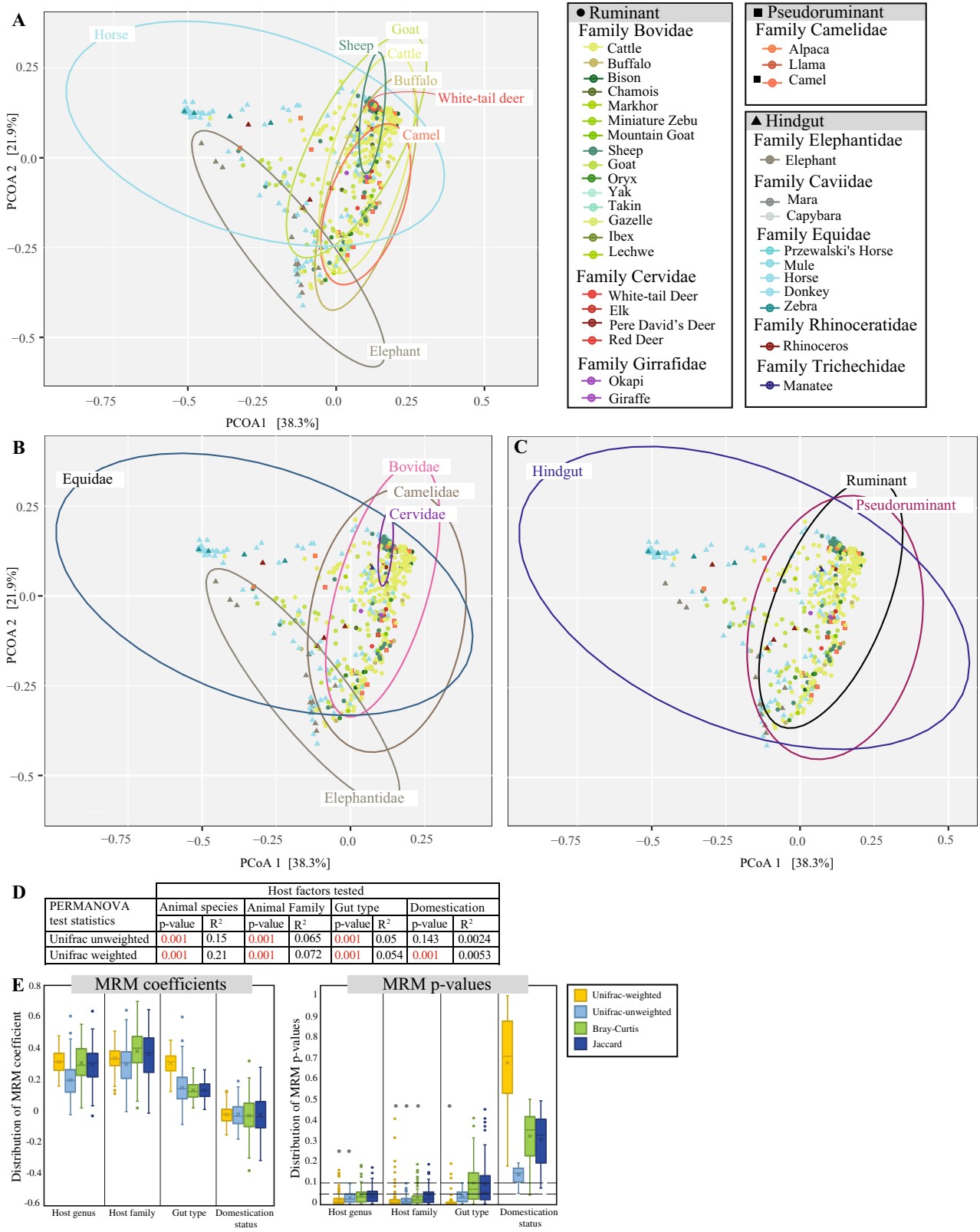

comparison-based methods: multiple regression of matrices (MRM), Mantel tests for matrices correlations, and Procrustes rotation[44,45], to confirm the role of host-related factors in shaping AGF community. Results of matrices correlation using each of the three methods, and regardless of the index used, confirmed the importance of animal host species, family, and gut type in explaining the AGF community structure (Fig. S9). Further, we permuted the MRM analysis (100 times),

where one individual per animal species was randomly selected for each permutation. Permutation analysis (Fig. 4e) yielded similar results to those obtained from the entire dataset (Fig. S9b), demonstrating that the obtained results are not affected by community composition variation among hosts of the same animal species.

Collectively, our results suggest a pattern of phylosymbiosis, with closely related host species harboring similar AGF communities[46]. To

**Fig. 4 | Patterns of AGF beta diversity. A–C** Principal coordinate analysis (PCoA) ordination plots based on AGF community structure in the 661 samples studied here. PCoA was constructed using the phylogenetic similarity-based weighted Unifrac index. The percent variance explained by the first two axes is displayed on the axes. Samples are color-coded by animal species, while the shape depicts the gut type as shown in the figure legend. Ellipses encompassing 95% of variance are shown for (**A**) animal species with >10 individuals (as labeled on the ellipses and as color-coded in the figure legend), (**B**) animal families with >10 individuals (as labeled on the ellipses), and (**C**) animal gut type (as labeled on the ellipses). **D** Results of PERMANOVA test for partitioning the dissimilarity among the sources of variation (including animal species, animal family, animal gut type, and animal lifestyle) for each of the phylogenetic similarity-based (unweighted and weighted Unifrac) indices used. The F statistic two-tailed *p*-value depicts the significance of the host factor in affecting the community structure, while the PERMANOVA statistic $R^2$ depicts the fraction of variance explained by each factor. **E** Results of MRM

analysis permutation (100 times, where one individual per animal species was randomly selected). Box and whisker plots are shown for the distribution of both the MRM coefficients (left) and the corresponding *p*-values (right) for the 100 permutations for each of the host factors (animal species, animal family, animal gut type, and animal lifestyle) and dissimilarity indices used (Unifrac weighted, Unifrac unweighted, Bray-Curtis, and Jaccard). *P*-values were obtained from the permutation test using the two-tailed pseudo-t method by ref. [109], and were not adjusted. If the *p*-value was significant (<0.05) in 75 or more permutations, the host factor was considered to significantly affect community structure (shown as an asterisk above the box and whisker plot). Boxplots extend from the first to the third quartile and the median is shown as a thick line in the middle. The whiskers extending on both ends represent variability outside the quartiles and are calculated as follows: Minimum whisker=minimum quartile−1.5 x inter-quartile range; Maximum whisker=maximum quartile+1.5xinter-quartile range. All points outside the box and whiskers are outliers.

confirm the significant association between the host animal and the AGF community, we employed PACo (Procrustes Application to Cophylogenetic) analysis with subsampling one individual per host species (*n* = 100 subsamples) and compared the distribution of PACo Procrustes residuals of the sum of squared differences within and between animal species (Fig. 5a), animal families (Fig. 5b), and animal gut types (Fig. 5c). Within each animal species, family, or gut type, PACo residuals varied minimally, indicating a high level of within host-AGF community association. Indeed, 90% of the residuals within animal species ranged from 0.0056 (buffalo) to 0.029 (elephant), within animal family ranged between 0.0048 (Giraffidae) to 0.029 (Elephantidae), and within gut type ranged between 0.007 (foregut) to 0.051 (hindgut) (boxplot heights in Fig. 5a–c). On the other hand, PACo residuals differed significantly between datasets (Wilcoxon two-sided adjusted *p*-value < 0.01) when animals belonged to different families, or different gut types (Fig. 5a–c, Table S6). These results indicated a strong cophylogenetic signal that was robust to intra-animal species microbiome variation.

### Identifying specific genus-host associations
Global phylogenetic signal statistics (Abouheif's Cmean, Moran's I, and Pagel's Lambda) identified 37 genera with significant correlations to the host phylogenetic tree (*p*-value < 0.05 with at least one statistic) (Table S7). In addition to global phylogenetic signal statistics, we calculated local indicator of phylogenetic association (LIPA) values for correlations between specific genera abundances and specific hosts. Of the above 37 genera, 34 showed significant associations with at least one animal host (LIPA values ≥0.2), with 17 showing strong associations (LIPA values ≥1) with specific animal species, and 10 showing strong associations (LIPA values ≥1) with certain animal families (Fig. 5d). A distinct pattern of strength of association was observed: All hindgut fermenters exhibited a strong association with a few AGF genera: horses, Przewalski's horses, and zebras with the genus *Khoyollomyces*, mules with the uncultured genus AL3, *Orpinomyces*, and *Caecomyces*, donkeys with *Piromyces*, elephants with *Piromyces*, *Caecomyces*, and *Orpinomyces*, rhinoceroses with NY20, manatees with NY54 and *Paucimyces*, and maras with NY1 and *Orpinomyces*. Members of the animal family Equidae mostly showed association with the phylogenetically related genera *Khoyollomyces* and the uncultured genus AL3, suggesting a broader family-level association between both host and fungal families (Fig. 5d, Table S8). On the other hand, a much smaller number of strong host-AGF associations were observed in ruminants (5/22 animal species: NY19 in bison, RH2 in oryx, AL8 in buffalo, NY9, SK3, and *Caecomyces* in yak, and *Neocallimastix* in elk) (Fig. 5d, Table S8). However, this lack of strong LIPA signal was countered by the identification of multiple intermediate and weak cophylogenetic signals (LIPA values 0.2-1, yellow in Fig. 5d) per animal species. It therefore appears that an ensemble of genera, rather than a single genus, is mostly responsible for the phylosymbiosis signal observed in

ruminants. Indeed, DPCoA ordination biplot showed a clear separation of the hindgut families Equidae, and Rhinocerotidae, from the ruminant families Bovidae, Cervidae, and Giraffidae, with the pseudoruminant family Camelidae occupying an intermediate position. This corroborates the patterns suggested by LIPA values, with 14 genera contributing to the foregut community as opposed to only 9 for hindgut fermenters (Fig. S10).

### Effect of other host factors on AGF diversity and community structure
In addition to host phylogeny and domestication status, additional factors could impact AGF diversity and community structure including biogeography, animal age, animal sex, as well as diet. However, the effect of these non-host-related factors could potentially be conflated when examined across different hosts. One way to avoid such conflation is to limit the analysis to the same animal species (e.g., examining the effect of biogeography on the AGF community structure using cattle samples only). Assessment of alpha diversity patterns (Table S9) indicated that biogeography had a significant effect on alpha diversity in cattle (with all indices, *p*-value < 0.03), horses (with 3 out of 4 indices, *p*-value < 0.04), but not in goats (with 3 out of 4 indices, *p*-value > 0.1), or sheep (with 3 out of 4 indices, *p*-value > 0.05). On the other hand, animal sex largely had no significant effect on alpha diversity (*p*-value > 0.05), while animal age only showed a significant effect on the alpha diversity of horses (with all indices, *p*-value < 0.03), goats (with all indices, *p*-value < 0.01), and sheep (with 2 out of 4 indices, *p*-value < 0.003), but not cattle.

Our analysis on the potential role of biogeography on AGF community structure indicated that the country of origin significantly explained 3.9% of variance in cattle (F-test *p*-value = 0.015), 10.2% of variances in horses (F-test *p*-value = 0.001), 20.6% of variances in goats (F-test *p*-value = 0.001), and 33.8% of variances in sheep (F-test *p*-value = 0.001) (Fig. S11A–D, M). Similarly, animal age significantly explained 6–18% (depending on the animal species) (Fig. S11E–H, M), and animal sex significantly explained 3.1–18.0% (depending on the animal species) (Fig. S11I–M) of variances in AGF community structure.

Diet could also play an important role in shaping AGF diversity as previously shown for bacterial community comparison in animal fecal samples[47]. However, the assessment of the impact of diet on AGF community structure using the current dataset is not ideal, given the fact that animal species with enough replicates (cattle, goats, sheep, and horses) originated from domesticated settings where animals were fed a highly similar diet. As well, exact documentation of diet in wild herbivores over a long time span is not feasible. A more targeted effort, in which diet is purposefully manipulated in a similar cohort(s) of specific animal species and monitored over a prolonged time frame, is needed to address such an issue.

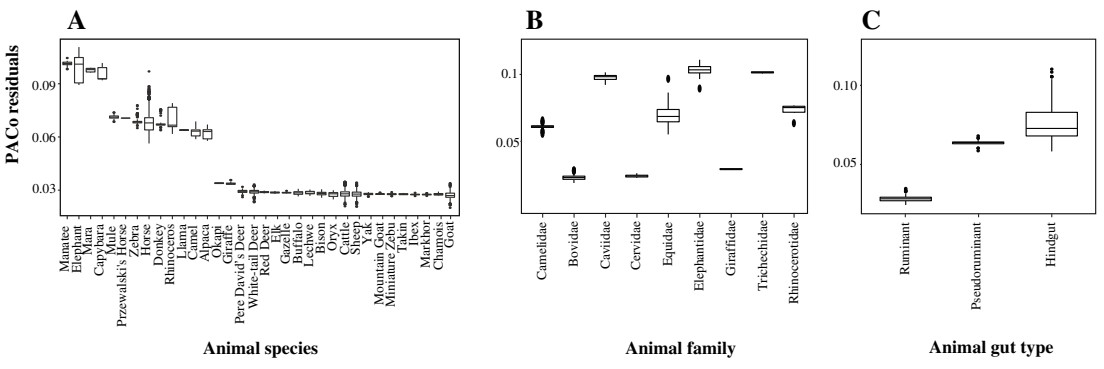

## Phylogenomic and molecular clock analyses correlate fungal-host preferences to co-evolutionary dynamics

The observed patterns of fungal-animal host preferences could reflect co-evolutionary symbiosis (i.e., a deep, intimate co-evolutionary process between animal hosts and AGF taxa). Alternatively, the observed patterns could represent a post-evolutionary environmental filtering process, where prevalent differences in in-situ conditions (e.g., pH, retention time, redox potential, feed chemistry) select for adapted taxa from the environment regardless of the partners' evolutionary history[48]. To address both possibilities, we generated new transcriptomic datasets for 20 AGF strains representing 13 genera and combined these with 32 previously published AGF transcriptomes[36–41].

We then used the expanded dataset (52 taxa, 14 genera) to resolve the evolutionary history of various AGF genera and estimate their divergence time. In general, most genera with a preference to hindgut fermenters occupied an early-diverging position in the Neocallimastigomycota tree, and a broad concordance between their estimated divergence estimate and that of their preferred host family was observed (Fig. 6). The genus *Khoyollomyces*, showing preference to horses and zebras (family Equidae), represented the deepest and earliest branching Neocallimastigomycota lineage, with a divergence time estimate of 67–50 Mya (Fig. 6). This estimate is in agreement with the divergence of the Equidae ~56 Mya[49,50]. As well, while the genera AL3 and NY54 are uncultured, and hence not included in the timing

**Fig. 5 | Phylosymbiosis patterns assessed using Procrustes Application to Cophylogenetic (PACo) analysis and Local Indicator of Phylogenetic Association (LIPA).** Distribution of PACo Procrustes residuals of the sum of squared differences within different animal species (**A**), animal families (**B**), and animal gut types (**C**). Boxplots extend from the first to the third quartile and the median is shown as a thick line in the middle. The whiskers extending on both ends represent variability outside the quartiles and are calculated as follows: Minimum whisker=minimum quartile−1.5 x inter-quartile range; Maximum whisker=maximum quartile +1.5xinter-quartile range. All points outside the box and whiskers are outliers. The number of data points used to calculate each box and whisker plot in (**A–B**) correspond to the number of samples belonging to each animal species, and animal family as defined in Fig. 1D. The number of data points used to calculate the box and whisker plot in (**C**) is shown on top of each plot. Results of two-sided Wilcoxon test for the significance of difference between PACo residuals are shown in Table S6 and

the significance asterisks are shown on top of the boxplots in **B**, **C**. **D** Local indicator of phylogenetic association (LIPA) values for correlations between genera abundances and specific hosts. The AGF tree on the left is a maximum likelihood midpoint rooted tree including only the 34 genera that were found to have significant associations with at least one animal host (LIPA values ≥0.2, p-value < 0.05). Bootstrap support is shown for nodes with >70% support. Average LIPA values for specific AGF genus-host genus associations (left) and AGF genus-host family association (right) are shown as a heatmap. Note that because average values are shown here, and due to the variation in the number of individuals belonging to each of the animal species, LIPA associations identified with animal species might not always be reflected at the family level. For example, AL3 and *Piromyces* are clearly associated with mules, and donkeys, respectively, but this association was not strong at the Equidae family level due to the small number of mules (n = 4) and donkeys (n = 5) studied compared to the total number of Equidae animals (n = 152).

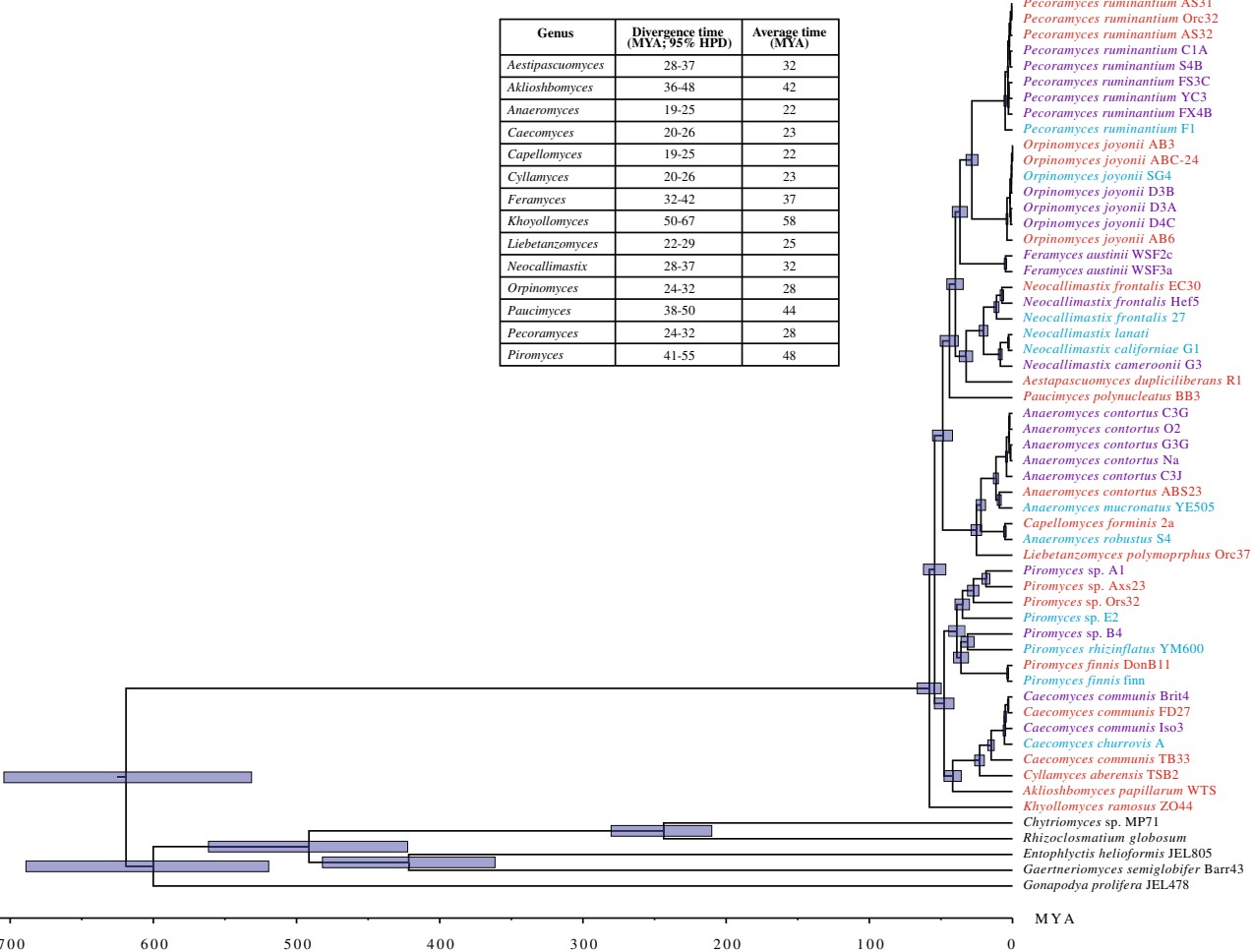

**Fig. 6 | Bayesian phylogenomic maximum clade credibility (MCC) tree of Neocallimastigomycota with estimated divergence time.** The isolate names are color coded to show data produced in this study (red), in previous studies by our group[36,37] (purple) and by other groups[38–41,80,110] (cyan). All clades above the rank of

the genus are fully supported by Bayesian posterior probabilities. The 95% highest-probability density (HPD) ranges (blue bars) are denoted on the nodes. For clarity, the average divergence time and 95% HPD ranges of each genus are summarized in the table.

analysis, their well-supported association with *Khoyollomyces* in LSU trees (Fig. 2a and S5) strongly suggests a similar early divergent origin. This is in agreement with the early evolution of the families of their hindgut preferred hosts: mules (family Equidae) for AL3, and manatee (family Trichechidae, evolving ~55 Mya)[50] for NY54. Similarly, the genus *Piromyces*, with a preference to elephants (family Elephantidae) and donkeys (Equidae), also evolved early (55-41 Mya), in accordance with the divergence time estimates for families Equidae and Elephantidae (~55 Mya)[49,50]. Finally, the early divergence time estimate of *Paucimyces*

(50-38 Mya) is again in agreement with its preference for the hindgut family Trichechidae (Manatee)[50].

Contrasting with the basal origins of AGF genera associated with hindgut fermenters, the majority of AGF genera showing strong, intermediate, or weak association with ruminants appear to have more recent evolutionary divergence time estimates (Fig. 6). These include many of the currently most abundant and ecologically successful genera, e.g., *Orpinomyces* (24-32 Mya), *Neocallimastix* (28-37 Mya), *Anaeromyces* (19-25 Mya), and *Cyllamyces* (20-26 Mya). These timings

are in agreement with estimates for the rapid diversification and evolution of the foregut fermenting high ruminant families Bovidae, Cervidae, and Giraffidae (18-23 Mya)[31,51], following the establishment and enlargement of the functional rumen[31]. While these results suggest the central role played by co-evolutionary phylosymbiosis in shaping the AGF community, timing estimates for a few AGF genera did not correspond to the evolutionary history of their preferred animal hosts. Such discourse patterns suggest a time-agnostic post-evolutionary environmental filtering process. The late-evolving genera *Orpinomyces* (24-32 Mya) and *Caecomyces* (20-26 Mya) (Fig. 6) were widely distributed and demonstrated intermediate and strong preferences not only to ruminants but also to hindgut fermenters (Fig. 5d), suggesting their capacity to colonize hindgut-fermenting hosts, the existence of which has preceded their own evolution. Collectively, these results argue for a major role of co-evolutionary phylosymbiosis and a minor role of post-evolutionary environmental filtering in shaping the AGF community in mammals.

### Evaluation of shotgun metagenomics for assessing AGF diversity and community structure in mammalian herbivorous

Shotgun metagenomic sequencing was conducted on a subset of samples ($n = 9$) to assess its utility in recovering AGF genomic fragments for comparative phylogenetic and functional analysis (Table S10). Within the entire 611 M reads obtained, only 1.56 M and 48 K reads were assigned to eukaryotic origins (using both Kaiju, and GOTTCHA, respectively). More importantly, reads assigned to AGF (38 K reads) were only encountered in one (horse) out of nine samples examined (Fig. S12).

To account for the possibility that identification of AGF at the Illumina reads level might be hampered by the short lead length and the low coding density in AGF genomes, reads were assembled into contigs using IDBA-UD (Table S11), and EukRep, a *k*-mer-based strategy for eukaryotic sequence identification in metagenomic datasets[52], was further used to identify contigs potentially belonging to eukaryotic organisms. Several contigs from each assembly were tentatively identified as potentially eukaryotic in origin by EukRep and flagged for further analysis. These contigs were subsequently binned using CONCOCT[53]. Additional analysis by filtering bins <1 M bp, and classification of larger bins using the prokaryotic classifier GTDB-tk[54]) demonstrated that these bins were indeed of a prokaryotic origin (Table S12). Further, none of the contigs had a GC content approaching the low GC values observed in sequenced fungal genomes (17–23%), again demonstrating their non-AGF origin (Table S12).

The observed low AGF recovery could be attributed to the lower proportion of fungal DNA in herbivorous feces when compared to bacteria, as previously described[55]. To confirm such observation in our samples, quantitative PCR (qPCR) on a subset of 40 samples (10 cattle, 10 goat, 10 sheep, and 10 horse samples) showed a relative 1:250,000 rRNA gene copy number between AGF and bacteria (Fig. S13).

## Discussion

Global amplicon-based, genomic, and metagenomic catalogs have significantly broadened our understanding of microbial diversity on Earth[56–60]. In this study, we generated and analyzed a global (661 samples, 34 animal species, 9 countries, and 6 continents) LSU amplicon dataset, as well as a comprehensive transcriptomic dataset (52 strains from 14 genera) for the Neocallimastigomycota. We focused on using this dataset for documenting the global scope of AGF diversity, as well as deciphering patterns and determinants of the herbivorous mycobiome. However, the size, coverage, and breadth of both datasets render them valuable resources for addressing additional questions and hypotheses by the scientific community.

Our study demonstrates that the scope of AGF diversity is much broader than previously suggested from prior efforts[27,61,62]. This broad expansion could be attributed to at least three factors: First, we examined previously unsampled and rarely sampled hosts, including manatee (a herbivorous marine mammal), mara, capybara, chamois, markhor, and takin. Indeed, a greater proportion of sequences belonging to novel genera were found in such samples (Fig. 2), and hence we posit that examining the yet-unsampled herbivorous mammals should be prioritized for novel AGF discovery. Second, we examined many replicates for several domesticated animal species and found that some novel genera were detected in some but not all samples from the same animal. Given the immense number of herbivores roaming the Earth, it is rational to anticipate that additional AGF diversity surveys of even well-sampled hosts could continue to yield additional novel lineages. Third, we accessed rare members of the AGF community through deep sequencing and found that 5 of the 56 novel genera were never identified in >0.1% abundance in any sample, and 16 of the 56 never exceeded 1% (Fig. 2g, S2b). The identification of sequences belonging to all these novel genera, including these perpetually rare ones (Fig. 2) strongly suggests their real occurrence and that such genera are not artifacts generated by primer bias, although such assertion could not be completely ruled out for the small fraction of genera that were identified in a few datasets and not confirmed by detection in SMRT datasets (e.g. NY38, and NY41 detected in 8, and 31 datasets, respectively, with abundances never exceeding 1%). The rationale for the existence, maintenance, and putative ecological role of rare members within a specific ecosystem has been highly debated[63]. We put forth two distinct, but not mutually exclusive, explanations for the maintenance of rare AGF taxa, both of which are based on prior assessments of the bacterial rare biosphere[63–66]. First, rare taxa could persist in nature by coupling slow growth rates to superior survival (e.g., high oxygen tolerance, formation of resistant structures outside the herbivorous gut), dispersal, and transmission capacities when compared to more abundant taxa[67]. Second, rare taxa could provide valuable ecological services under specific conditions not adequately captured by the current sampling scheme[68], e.g., specialization in attacking specific minor components in the animal's diet, superior growth in specific cases of gut dysbiosis, or during early stages of their hosts life. In newborn animals, the undeveloped nature of the alimentary tract[69], the liquid food intake, and distinct behavior, e.g. coprophagy in foals, may select for a distinct microbiome, and rare AGF members of the community could hence represent remnants of the community developing during the early days of the host life. Detailed analysis of the effect of dysbiosis on AGF communities, as well as the temporal development patterns from birth to maturity, is needed to experimentally assess the plausibility of both scenarios.

Our results highlight the importance of the hitherto unrecognized role of stochastic processes (drift and homogenizing dispersal) in shaping the AGF community in herbivores. The contribution of these processes was on par with (in the hindgut fermenting and pseudoruminant families) or exceeding (in the ruminant families Bovidae and Cervidae) that of deterministic niche-based processes (Fig. 3i). We attribute the high contribution of drift to the restricted habitat and small population sizes of AGF in the herbivorous gut, conditions known to elicit high levels of drift[35]. As well, the highly defined functional role for AGF in the herbivorous gut (initial attack and colonization of plant biomass), high levels of similarity in metabolic abilities, substrates preferences, and physiological optima across genera argue for a null-model scenario, where the roles of phylogenetically distinct taxa are ecologically interchangeable. The importance of homogenizing dispersal (Fig. 3i) suggests a high and efficient dispersal rate leading to community homogenization. While the strict anaerobic nature of AGF could argue that dispersal limitation, rather than homogenizing dispersal, should be more important in shaping AGF communities. However, such a perceived transmission barrier could plausibly be surmounted via direct vertical mother-to-offspring transfer by post-birth grooming, as well as direct horizontal

transmission between animals, or through feed-fecal cross contamination in close quarters[70].

A greater level of stochasticity was observed in ruminants compared to hindgut fermenters. This could be due to the proximity of the prominent AGF-harboring chamber (rumen) to the site of entry (mouth) in ruminants[71,72], compared to the distant location of the reciprocal chamber (caecum) in hindgut fermenters[73,74]. This proximity could result in a greater rate of secondary airborne transmission in foregut fermenters, as well as a greater level of selection for AGF inoculum in hindgut fermenters during their passage through the alimentary tracts (with various lengths and residence times). The observed pattern could also be due to the high-density rearing conditions and a higher level of inter-species cohabitation between many ruminants (e.g. cattle, sheep, goats)[75,76], as opposed to the relatively lower density and cross-species cohabitation for hindgut fermenters (e.g. horses, elephants, manatees)[77].

While stochastic processes play a role in AGF community assembly, the role of deterministic processes remains substantial (Fig. 3). Host-associated factors are logical factors to examine as key drivers of AGF community structure. Differences in overall architecture, size, and residence time in alimentary tracts of different hosts could result in niche-driven selection of distinct AGF communities. In addition, variation in bacterial and archaeal community structures between hosts could also elicit various levels of synergistic, antagonistic, or mutualistic relationships that impact the AGF community[78–81]. However, domestication status could counter, modulate, or override host identity. Domesticated animals are fed regularly and frequently a monotonous diet, compared to the more sporadic feeding frequency and more diverse feed types experienced by non-domesticated animals. Such differences could select for AGF strains suited for each lifestyle. Furthermore, the close physical proximity and high density of animals in domesticated settings are conducive to secondary airborne transmission, while the more dispersed lifestyle of wild herbivores could elicit a more stable community within a single animal species.

Ordination clustering patterns (Fig. 4, S8) and PERMANOVA analysis demonstrated that host-associated factors explained a much higher proportion of the observed variance, when compared to host's domestication status (Fig. 4d). All hindgut fermenters exhibited strong associations with a few AGF genera, while multiple intermediate cophylogenetic signals were identified for foregut fermenters (Fig. 5d). This suggests that enrichments of an ensemble of multiple genera, rather than a single genus, is mostly responsible for the distinct community structure observed in foregut fermenters. These patterns of strong animal-host correlation correspond to the patterns of lower stochasticity (Fig. 3), and lower alpha diversity (Fig. S7) observed in hindgut fermenters.

As described above, the predicted role of phylosymbiosis in shaping AGF community structure in extant animal hosts could reflect two distinct, but not mutually exclusive, mechanisms; co-evolutionary phylosymbiosis, and post-evolutionary host filtering. Phylogenomic approaches using whole genomic and/or transcriptomic datasets are a promising tool for resolving such relationships[82–86]. Our results from transcriptomics-enabled phylogenomic and molecular clock analysis indicate a more prevalent role for co-evolutionary phylosymbiosis in shaping the observed pattern of AGF diversity. Specifically, it appears that the evolution of various herbivorous mammalian families, genera, and species following the K-Pg extinction event and continuing through the early Miocene, and the associated evolutionary innovations in alimentary tract architecture (e.g., the evolution of the three-chambered forestomach of pseudoruminants, and the four-chambered stomach of ruminants), was associated with a parallel evolutionary diversification process within the Neocallimastigomycota. This is supported by the preference of earliest divergent AGF genera to hindgut fermenting hosts, e.g. *Khoyollomyces* and associated genera (AL3 and NY54) to members of the Equidae[8,87,88], as well as the

general basal position of additional hindgut-preferring genera, e.g., *Piromyces* (41-55 Mya) and *Paucimyces* (38-50 Mya). This is in agreement with the fact that early mammals roaming the Earth past the K-Pg boundary (~65.5 Mya) were hindgut fermenters. On the other hand, the recent origin for the foregut-preferring genera *Orpinomyces*, *Neocallimastix*, and *Anaeromyces* (22-32 Mya) suggests this followed the earlier evolution (~ 40 Mya) of a functional and enlarged rumen[31], and the subsequent rapid diversification and evolution of multiple families in the high ruminants (Suborder Ruminantia, Infraorder Pecora), e.g. Bovidae, Cervidae, Giraffidae (18-23 Mya)[31,51]. As such, organismal and gut evolution appears to have provided novel niches that putatively facilitated rapid AGF genus-level diversification in the early Miocene. However, in addition to phylosymbiosis, post-evolutionary host filtering also appears to play a role in shaping the AGF community. For example, members of the genus *Orpinomyces* showed a strong association with a wide range of animal families and gut types (Fig. 5d). The reason for the ecological success of *Orpinomyces* in multiple hosts is currently uncertain, but members of this genus exhibit robust polycentric growth pattern, enabling fast vegetative production via hyphal growth and fragmentation.

Finally, it could be argued that, compared to amplicon datasets, the use of shotgun metagenomics provides a more detailed assessment of the patterns and determinants of AGF diversity, as it can provide both phylogenetic and functional information. However, our metagenomic shotgun sequencing on a subset of samples failed to recover AGF-affiliated sequences from such samples (Table S12, Fig. S12). This could be attributed to extremely low AGF DNA levels when compared to bacterial DNA, as we confirm using qPCR in a subset of our samples (Fig. S13), as well as to the low coding density, extremely high AT content, and proliferation of repeats in AGF genomes, which may have hindered the assembly of reads[39,40,89]. The absence of AGF-affiliated sequences has also been observed in prior metagenomic studies conducted on fecal samples of herbivores (Table S13). Enrichment efforts, e.g., using antibiotics to suppress bacterial growth, could potentially enrich AGF from fecal samples, but such effort results in the enrichment of an AGF monoculture[90], nullifying its utility for diversity assessment and documentation. As well, it is important to note that, currently, only genomes representative of 5 of the 20 cultured AGF genera, and none from uncultured genera are publicly available[39–41,89], which greatly prevents accurate phylogenetic anchoring of any fragments that might have been recovered from metagenomic efforts.

In summary, our results demonstrate that the scope of fungal diversity in the herbivorous gut is much broader than previously implied from prior culture-dependent, culture-independent, and –omics surveys[27,39,91–93], quantify the relative contribution of various ecological factors in shaping AGF community assembly across various hosts, and demonstrate that host-specific evolutionary processes (e.g. evolution of host families, genera, and gut architecture) played a key role in driving a parallel process of AGF evolution and diversification.

## Methods
### Study design and sample selection
A total of 661 fecal samples belonging to 34 different mammalian animal species and 9 families of ruminant, pseudoruminant, and hindgut fermenters were included in the final analysis (Fig. 1b, d, Supplementary Data 1). Samples were obtained from 15 different research groups using a single standardized procedure (Supplementary methods). Capturing the overall community in animals gut using fecal samples is a widely used approach[51,94–96]. Importantly, fecal samples allows the inclusion of animals with different GIT architectures and different sites for the majority of fermentation in the same study (foregut ruminants, pseudoruminant foregut fermenters, and hindgut fermenters, see Fig. 1a). Fecal

sampling also circumvents logistical and ethical issues associated with sampling animals' internal structures (e.g., euthanasia, fistulation, or gastric tube insertion to obtain rumen and diverticula samples from ruminants and pseudoruminants) and overcomes the difficulty of acquisition of such samples from wild animals or precious zoo-housed animals. Further, fecal samples can be easily homogenized, which is in stark contrast to the physical heterogenicity of rumen samples, for example, where communities could greatly differ between rumen fraction type (liquid vs solid)[97], hence complicating reproducibility.

## DNA extraction

DNA extractions from fecal samples were conducted in eight laboratories using DNeasy Plant Pro Kit (Qiagen®, Germantown, Maryland, USA) according to manufacturer's instructions. The kit has previously been evaluated by multiple laboratories in prior surveys of AGF diversity[27,98].

## Illumina sequencing

All PCR amplification reactions, amplicon clean-up, quantification, index and adaptor ligation, and pooling were conducted in a single laboratory (Oklahoma State University, Stillwater, OK, USA) to eliminate inter-laboratory variability. All reactions utilized the DreamTaq Green PCR Master Mix (ThermoFisher, Waltham, Massachusetts, USA), and AGF-LSU-EnvS primer pair (AGF-LSU-EnvS For: 5'-GCGTTTRRCA CCASTGTTGTT-3', AGF-LSU-EnvS Rev: 5'-GTCAACATCCTAAGYG TAGGTA-3')[98] targeting a ~370 bp region of the LSU rRNA gene (corresponding to the D2 domain), an amplicon size enabling high throughput sequencing using the Illumina MiSeq platform. Recent work has established the superiority of LSU, over ITS1 commonly used in other fungal lineages due to its length homogeneity, and lack of intragenomic variability[27,98]. Negative (reagents only) controls were included in all PCR amplifications to detect possible cross-contamination. Pooled libraries (300-350 samples) were sequenced at the University of Oklahoma Clinical Genomics Facility (Oklahoma City, OK, USA) using the Illumina MiSeq platform (supplementary methods).

## Complementary PacBio sequencing

As a complementary approach to Illumina sequencing, we conducted PacBio sequencing on a subset ($n = 61$) of the Illumina-sequenced samples to amplify the D1/D2 LSU region (~700 bp). Primers utilized were the fungal forward primer (NL1: 5'- GCATATCAATAAGCGGA GGAAAAG-3'), and the AGF-specific reverse primer (GG-NL4: 5'-TCAA CATCCTAAGCGTAGGTA-3')[27,99]. Details on the rationale for PacBio sequencing, as well as PCR amplification, amplicon clean-up, quantification, index and adaptor ligation, and pooling are in the supplementary methods.

## Sequence processing, and taxonomic and phylogenetic assignments

Protocols for read assembly, and sequence quality trimming, as well as procedures for calculating thresholds for species and genus delineation and genus-level assignments are provided in supplementary methods. Briefly, pairwise sequence divergence estimates comparison between SMRT and Illumina amplicons showed very high correlation ($R^2 = 0.885$, Fig. S14), and indicated that the 2% sequence divergence cutoff previously proposed as the threshold for delineating AGF species using the D1/D2 region (based on comparisons of validly described species)[29] is equivalent to 3.5% using the D2 region only, and the 3% sequence divergence cutoff previously proposed as the threshold for delineating AGF genera using the D1/D2 region[29] is equivalent to 5.1% using the D2 region only (Fig. S14). Assignment of sequences to AGF genera was conducted using a two-tier approach for genus-level phylogenetic placement as described previously[27,29], and as detailed in the supplementary methods.

## Role of stochastic versus deterministic processes in shaping AGF community assembly

We assessed the contribution of various deterministic and stochastic processes to the AGF community assembly using both normalized stochasticity ratio (NST)[33], and the null-model-based quantitative framework implemented by refs. 34,35. The NST ratio infers ecological stochasticity, however, values do not pinpoint the sources of selection (determinism) or stochasticity. Also, NST values are calculated solely based on taxonomic diversity indices with no consideration of the phylogenetic turnover in the community. To quantify the contribution of various deterministic (homogenous and heterogenous selection) and stochastic (dispersal preference, limitation, drift) processes in shaping the AGF community assembly, we used a two-step null-model-based quantitative framework that makes use of both taxonomic ($RC_{Bray}$) and phylogenetic ($\beta NRI$) β-diversity metrics[34,35] (supplementary methods).

## Factors impacting AGF diversity and community structure

We considered two types of factors that could potentially impact AGF diversity and community structure: host-associated factors, and non-host-associated factors. For host-associated factors, we considered animal species, animal family, and animal gut type, while for non-host-associated factors we considered domestication status, biogeography (country of origin), age, and sex. For testing the effect of biogeography, age, and sex, we carried out comparisons only on samples belonging to the same animal species to control for other host-associated factors. For these comparisons, only the four mostly sampled animal species (cattle, goats, sheep, and horses) were considered.

Alpha diversity estimates were calculated as described in the supplementary document. Beta diversity indices (phylogenetic similarity-based e.g., unweighted and weighted Unifrac) were calculated using the ordinate command in the phyloseq R package. The pairwise values were used to construct ordination plots (both PCoA and NMDS) using the function plot_ordination in the phyloseq R package. RDA plots were also constructed using the genera center log-ratio transformed abundance data. To partition the dissimilarity among the sources of variation (including animal host species, animal host family, animal gut type, and animal lifestyle), PERMANOVA tests were run for each of the above beta diversity measures using the vegan command adonis, and the F-statistics p-values were compared to identify the host factors that significantly affect the AGF community structure. The percentage variance explained by each factor was calculated as the percentage of the sum of squares of each factor to the total sum of squares.

To further quantitatively assess factors that explain AGF diversity, we used three additional multivariate regression approaches based on matrices comparison: multiple regression of matrices (MRM), Mantel tests for matrices correlations, and Procrustes rotation. Bray-Curtis, Jaccard dissimilarity, Unifrac weighted, and Unifrac unweighted dissimilarity matrices were compared to a matrix of each of the host factors tested (animal host species, animal host family, animal gut type, and animal lifestyle) using the commands MRM, and mantel in the ecodist R package, for running multiple regression on matrices, and Mantel tests. The Procrustes rotation was calculated using the protest command in the vegan R package. The significance, and importance of the host factor in explaining the AGF community structure were assessed by comparing the p-values, and coefficients ($R^2$ regression coefficients of the MRM analysis, Spearman correlation coefficients of the Mantel test, and symmetric orthogonal Procrustes statistic of the Procrustes analysis), respectively. Finally, to assess the sensitivity of multivariate regression methods to community composition variation among hosts of the same species, we permuted the MRM analysis 100 times, where one individual per animal species was randomly selected. For each of these permutations, and for each dissimilarity matrix-host factor comparison, a p-value and an $R^2$

regression coefficient were obtained. We considered a host factor significant in explaining AGF community structure if in the permutation analysis, the *p*-value obtained was significant ($p < 0.05$) in at least 75 permutations (supplementary methods).

## Assessing phylosymbiosis patterns

To test for patterns of phylosymbiosis, and the presence of a cophylogenetic signal between the animal host and the AGF genera constituting the gut community, we used Procrustes Application to Cophylogenetic Analysis (PACo) through the paco R package (supplementary methods). For pinpointing specific animal host-fungal associations, we employed two approaches. We first used the phyloSignal command in the phylosignal R package to calculate three global phylogenetic signal statistics, Abouheif's Cmean, Moran's I, and Pagel's Lambda. The values of these statistics plus the associated p-values were employed to identify the AGF genera that have a significant association with an animal host. We then used the lipaMoran command in the phylosignal R package to calculate LIPA (Local Indicator of Phylogenetic Association) values for each sample-AGF genus pair, along with the associated p-values of association significance. For AGF genera showing significant associations (LIPA *p*-values < 0.05), we calculated average LIPA values for each animal host species, and animal family. We considered average LIPA values in the range of 0.2–0.4 to represent weak associations, in the range 0.4–1 to represent moderate associations, and above 1 to represent strong associations. To further explore the notion that enrichments of an ensemble of multiple genera, rather than a single genus, is responsible for the distinct community structure observed in ruminants and pseudoruminants, we used the ordinate command in phyloseq followed by plot_ordination to construct a double principal coordinate analysis (DPCoA) plot.

## Transcriptomic analysis

Transcriptomic analysis was conducted to obtain gene content from a broad range of AGF taxa rather than assessing expression patterns of a single or few species. Use of transcriptomic rather than genomic datasets was driven by the relative ease of obtaining AGF transcriptomes compared to genomes and the ready generation of coding sequences with no introns from transcriptomic data. Prior studies by our research group have generated 21 transcriptomes from 7 genera[36,37]. Here, we added 20 transcriptomes from 7 additional genera, isolated during a long-term multi-year isolation effort in the authors' laboratory[27,29], and included an extra 11 publicly available transcriptomic datasets[38–41]. The dataset of 52 transcriptomes was used for phylogenomic analysis as described in ref. 42. For RNA extraction, cultures grown in rumen fluid-cellobiose medium[100] were vacuum filtered then grounded with a pestle under liquid nitrogen. Total RNA was extracted using Epicentre MasterPure yeast RNA purification kit (Epicentre, Madison, WI, USA) according to manufacturer's instructions. Transcriptomic sequencing using Illumina HiSeq2500 platform and 2 × 150 bp paired-end library was conducted using the services of a commercial provider (Novogene Corporation, Beijing, China), or at the Oklahoma State University Genomics and Proteomics Center. The RNA-seq data were quality trimmed and de novo assembled with Trinity (v2.6.6) using default parameters. Redundant transcripts were clustered using CD-HIT[101] with identity parameter of 95% (−c 0.95), and subsequently used for peptide and coding sequence prediction using the TransDecoder (v5.0.2) (https://github.com/TransDecoder/TransDecoder) with a minimum peptide length of 100 amino acids. BUSCO[102] was used to assess transcriptome completeness using the fungi_odb10 dataset modified to remove 155 mitochondrial protein families as previously suggested[38]. In addition, five Chytridiomycota Genomes (*Chytriomyces* sp. strain MP 71, *Entophlyctis helioformis* JEL805, *Gaertneriomyces semiglobifer* Barr 43, *Gonapodya prolifera* JEL478, and *Rhizoclosmatium globosum* JEL800) were included to

provide calibration points. The same phylogenomic dataset (670 protein-coding genes) produced by Hanafy et al.[42] was used as the original input. Gap regions were removed using trimAl v1.4[103]. Alignment files that contained no missing taxa and were longer than 150 nucleotide sites were selected for subsequent analyses. By employing a greedy search in PartitionFinder v2.1.1[104], the 88 selected alignments were grouped into 15 partitions with independent substitution models. All partition files and respective models were loaded in BEAUti v1.10.4[105] with calibration priors specified as previously described[37] ((i) a direct fossil record of Chytridiomycota from the Rhynie Chert (407 Mya) & (ii) the emergence time of Chytridiomycota (573 to 770 Mya as 95% HPD)) for Bayesian inference and divergence time estimation implemented in BEAST v1.10.4. The Birth-Death incomplete sampling tree model was employed for interspecies relationship analyses. Unlinked strict clock models were used for each partition independently. Three independent runs were performed for 50 million generations and Tracer v1.7.1[106] was used to confirm that sufficient effective sample size (ESS > 200) was reached after the default burn-in (10%). The maximum clade credibility (MCC) tree was compiled using TreeAnnotator v1.10.4[105].

## Shotgun metagenomic sequencing, read processing, and assembly

Shotgun metagenomic sequencing was conducted on a subset of samples ($n = 9$) from a cow ($n = 1$), horse ($n = 1$), goat ($n = 1$), sheep ($n = 1$), white-tail deer ($n = 1$), bison ($n = 1$), buffalo ($n = 1$), camel ($n = 1$), and elephant ($n = 1$) using the services of the University of Oklahoma Clinical Genomics Facility. Details on the library preparation and sequencing platform are shown in the supplementary methods. Reads were quality-trimmed using trimmomatic and assembled using IDBA-UD. Contigs with potential eukaryotic origin were identified using EukRep, followed by their binning using CONCOCT. Details on read processing, assembly, and eukaryotic binning are shown in Fig. S15 and detailed in the in supplementary methods.

## Quantitative PCR (qPCR)

We quantified and compared total AGF to total bacterial load in a subset of the samples from ten cattle, ten goats, ten sheep, and ten horses using quantitative PCR. This was done to further confirm the extremely low relative abundance of AGF DNA when compared to bacterial DNA in the herbivorous feces. The same primer pair (AGF-LSU-EnvS and AGF-LSU-EnvS Rev) used in the amplicon-based diversity survey described above was also used for qPCR quantification. The bacterial load was quantified in the same 40 samples using the 515 F and 806 R prokaryotic-specific primer pair amplifying the 16S rRNA V4 hypervariable region[107]. Detailed methodological analyses are presented in the Supplementary methods, and results are presented as rRNA copies/g sample.

## Reporting summary

Further information on research design is available in the Nature Portfolio Reporting Summary linked to this article.

## Data availability

Illumina amplicon reads generated in this study have been deposited in GenBank under BioProject accession number PRJNA887424, BioSample accession numbers SAMN31166910- SAMN31167478, and SRA accessions SRR21816543-SRR21817111. PacBio sequences were deposited in GenBank as a Targeted Locus Study project under the accession KFWW00000000. The version described in this paper is KFWW01000000. PacBio sequence representatives of the 49 novel AGF groups were deposited in GenBank under accession numbers OP253711-OP253963. Raw Illumina RNA-seq read sequences are deposited in GenBank under the BioProject accession number PRJNA847081, BioSample accession numbers SAMN28920465-

SAMN28920484, and individual SRA accessions SRR19612694-SRR19612713. Metagenomic reads were deposited in GenBank under BioProject accession number PRJNA887424, BioSample accession numbers SAMN34141577- SAMN34141585, and SRA accessions SRR24145987- SRR24145995.

## Code availability

Code for phylogenomic analysis is available at https://github.com/stajichlab/PHYling_unified. Code used to create all other figures is available at https://github.com/nohayoussef/AGF_Mammalian_Herbivores [108].

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

## Acknowledgements

This work has been supported by the NSF grant number 2029478 to MSE and NHY. Computational resources used at Oklahoma State University was supported by NSF grant OAC-1531128. Additional support has been provided by the New Zealand Ministry of Business, Innovation and Employment Strategic Science Investment Fund AgResearch Microbiomes programme; Fondo di Ateneo per la Ricerca 2020 University of Sassari and by funds obtained from Fondazione di Sardegna, Italy, FDS2223MONIELLO-CUP J83C22000160007 (GM); HiPoAF project funded by the Austrian Science Fund (FWF) grant number I3808 (M.N., J.M.V. and S.M.P), and Deutsche Forschungsgemeinschaft (DFG) as part of the project, HiPoAF-D.A.CH project number LE 3744/4-1 (DY). We thank Kelle Sundstrom, Emily Looper, and Ruth Scimeca (Oklahoma State University College of Veterinary Medicine), Jennifer D'Agostino, and Rebecca Snyder (Oklahoma City Zoo), as well as multiple ranchers and farmers throughout the USA for collecting and providing samples. We also thank David Hume, Katherine Lowe, Richard Muirhead, Christian Sauermann, Shengjing Shi, and Gosia Zobel who contributed the range of New Zealand samples. We also thank the Alpenzoo Innsbruck (Austria) for providing samples and help during sampling. We also thank ZOO Garden Ostrava, Farm Tehov, and Farm Duběnka Ostrava (Czech Republic) for help with sample collections. We also thank Katelyn Pixley, Ryzzah Trinidad, and Russell Croy for technical assistance.

## Author contributions

Conceptualization and experimental design by N.H.Y. and M.S.E., Sample collection, archiving, and laboratory experimentation by C.H.M., A.L.J., A.X.A., J.H., C.J.P., R.A.H., A.S.Y., M.A.T., C.D.M., P.H.J., M.S., P.R., M.N., J.M.V., S.M.P., A.L.G., J.H., D.Y., K.F., D.J.G., R.V., G.M., S.M., M.T.K., Y.I.N., S.S.D., and A.P.F. Data analysis by C.H.M., Y.W., J.E.S., J.A.E., N.H.Y., and M.S.E. Manuscript writing by N.H.Y. and M.S.E. Funding acquisition by C.D.M., P.H.J., G.M., M.N., J.M.V., S.M.P., D.Y., N.H.Y., and M.S.E.

## Competing interests

The authors declare no competing interests.

## Additional information

Noha H. Youssef or Mostafa S. Elshahed.

[1]Oklahoma State University, Department of Microbiology and Molecular Genetics, Stillwater, OK, USA. [2]Department of Biological Sciences, University of Toronto Scarborough, Toronto, ON, Canada. [3]Department of Microbiology and Immunology, Faculty of Pharmacy, Cairo University, Cairo, Egypt. [4]AgResearch Ltd, Grasslands Research Centre, Palmerston North, New Zealand. [5]Department of Applied Microbiology and Food Technology, Research Institute for Bioscience and Biotechnology (RIBB), Kathmandu, Nepal. [6]Universität Innsbruck, Faculty of Biology, Department of Microbiology, Innsbruck, Austria. [7]Department of Microbiology and Plant Pathology, University of California, Riverside, Riverside, CA, USA. [8]Langston University, Langston, OK, USA. [9]Bavarian State Research Center for Agriculture, Freising, Germany. [10]Institute of Animal Physiology and Genetics Czech Academy of Sciences, Prague, Czechia. [11]Área de Microbiología, Facultad de Ciencias Médicas, Universidad Nacional de Cuyo, Mendoza, Argentina. [12]Prague Zoo, Prague, Czechia. [13]Department of Veterinary Medicine, University of Sassari, Sardinia, Italy. [14]University of Milan, Dept. of Agricultural and Environmental Sciences, Milan, Italy. [15]Anaerobic Fungi Network, Kerkdriel, Netherlands. [16]Bioenergy Group, Agharkar Research Institute, Pune, India. [17]Oklahoma State University, Department of Animal and Food Sciences, Stillwater, OK, USA. ✉e-mail: noha@okstate.edu; mostafa@okstate.edu

