## [Peer Review File · Nature Communications]

Reviewer comments, first round

Reviewer #1 (Remarks to the Author):

This large metabarcoding study investigates the poorly explored communities of anaerobic gut fungi living in association with herbivorous mammals (the herbivore mycobiome). The host sampling scheme is quite large, encompassing herbivores from across the globe (although concentrated in the United States), spanning significant taxonomic breadth, and falling into three major categories (i.e., foregut-fermenting ruminants, foregut-fermenting pseudoruminants, and hindgut fermenters). Noted by the authors, the taxonomic sampling scheme associated with this study includes some never- or rarely sampled herbivore hosts. The authors identify at least 56 novel genera and 4 novel families, dramatically increasing the depth of Neocallimastigomycota sampling. They estimate the impact of stochastic and deterministic effects on driving differences in mycobiome structure, finding that stochastic effects (e.g., drift) play a larger role than previously appreciated and that, interestingly, domestication of herbivores has little apparent effect on mycobiome assemblage. Using a time-calibrated tree based on transcriptome data in conjunction with known diversification times for herbivorous hosts, the authors suggest that coevolution of gut fungi with their hosts has played a more important role than host-filtering in shaping present day gut communities.

This manuscript chronicles a significant amount of work and presents findings that will be interesting to a diverse biological audience, whether they be centered predominantly in mycology, zoology, or elsewhere. The manuscript is well-written (save for some minor typos) and the analyses conducted are appropriate and sufficient toward asserting the conclusions it draws. The authors should be commended for the lengths they have gone to to verify their results, including the use of multiple primer pairs and sequencing platforms as well as the use of dynamic cutoffs for genus level delineations. Given the density of data and breadth of important insights described here, I think this manuscript will find a well-deserved home in Nature Communications. At a finer level, I think there are some points that could be better clarified and some changes that should be made to several figures before that can happen.

Major comments and suggestions

1. Given the broad audience of Nature Communications, I think the article could benefit from a more basic and concise description of the different types of herbivore guts delineated in the manuscript (as it is a central concept). I am not an expert in this area and found myself going back to the introduction multiple times to check the meanings of terms, identify which type of gut different animal groups had, etc. I think that this suggestion could be effectively addressed with a figure that diagrams the major anatomical differences between different gut types and places them in an evolutionary perspective (e.g., a small diagrammatic tree of when they evolved and which animal groups have which gut type). I think this small step to bring readers naive to herbivore gut anatomy (such as myself) will make it easier for the authors to get their exciting conclusions across downstream.

2. Throughout the manuscript, the finding that a portion of novel genera were only ever found at low abundances ($5/56 \leq 0.1\%$ and $15/56 \leq 1\%$) is used to make the point that these taxa are perpetually rare. The authors make this assertion stronger by their having factored out the impact of primer bias (by using multiple primer pairs), sequencing technology (by using Illumina and PacBio chemistry), and sequencing libraries are sufficient depth. While I commend the authors for their attention to conducting meaningful controls and replication to this end, I think that the impact of (especially) primer bias, cannot be completely ruled out and should be mentioned more often as a possible driver of results. PCR always introduces bias and these primer pairs were necessarily designed agnostic to the many new taxa that they discover in this study. It is not surprising then that many of these novel genera are often present at such low abundances. Can the authors please either (i) clarify how they have sufficiently mitigated primer bias to the point where it can be ruled out (ii) add text when mentioning this assertion of rarity that clarifies the potential impact of primer bias (e.g., L164-167, L347-348, etc.). As a minor point in the same

vein, I would caution the authors against fully synonymizing read abundance with true abundance.

3. I think the coloring and organization of some of the figures could be improved toward concise communication of the conclusions.

- Figure 1C: There are color pairs in the rightmost legend (accompanying the stacked bar plot) that are going to be imperceptible to readers – there are just too many. I would suggest either (i) summing genera below some percentage into a single category or (ii) if you must keep each genus separate, order the legend such that it matches the stacked bar from high to low proportion. Also, the text alignment overlaid the stacked bar needs to be tightened up – on my PDF some are overlapping or misaligned with each other.
- Figure 1D: On the right bar charts, some background shading (e.g., alternative grey and white) between different fungal taxa (in addition to the bar coloration) would help the reader delineate which bars correspond to which taxon. It's not difficult when the bar coloration is visible, but in many cases it is not (e.g, *Joblinomyces* – *Buwchfawrowmyces*).
- Figure 3A-H: Significant comparisons should be marked here and p-values could also be shown on the plots.
- Figure 3I: Could use a y-axis label, despite it somewhat obviously being percentage.
- Figure 4A: Could the points on the NMDS/PCoA plots be smaller? At present, the degree of overlap makes it is hard to visually see when points more or less clustered along the axes.
- Figure 5D: It looks like some color legends are missing here.

Minor comments and suggestions

L171. Unmatched parentheses

L171. 'formed additional' should be 'formed an additional'

L171-174. The authors might consider using the same sentence structure here (new families) as in the sentence above (accepted families) instead of the long piece of text in parentheses.

L370. I'd rewrite to: 'where the roles of phylogenetically distinct taxa are ecologically interchangeable.'

L374. 'community' should be 'communities'

L374. 'readily surmounted' is strong language – are the described processes (e.g. post-birth grooming) that effective are circumventing this barrier?

L388. 'factor' should be 'factors'

L455. 'explaining' should be 'explained'

L454-457. incomplete sentence as currently written

Supp-L452. I'm not sure if this has to do with how I downloaded my particular version of this file, but Figure S4 is of poor quality (pixelated).

Reviewer #2 (Remarks to the Author):

The authors profiled the mycobiome of 661 fecal samples (sufficient sample size) from multiple herbivorous species, which further expanded the research scope of anaerobic gut fungi and increased our understanding of the diversity of gut mycobiome. The biological question raised here

is essential. The bioinformatic analyses are logical, and the data interpretations are sufficient enough. However, I still have several major and minor concerns about the current study that need to be further clarified and addressed.

Major concerns:

1. For ruminal species included in this study, I don't think the fecal mycobiome could represent the actual function of the gut mycobiome inhabiting these hosts; most of the fungi play functions in the rumen instead of the hindgut. In this case, why did the authors choose fecal samples?
2. DNeasy Plant Pro Kit (Qiagen®, Germantown, Maryland) was applied to extract fungi DNA from fecal samples. Is this a validated protocol for isolating fungi DNA? Are there any previous studies testing the efficiency and accuracy of this kit in terms of fungi DNA isolation?
3. The authors chose LSU rRNA gene as the targeted region. If this is only for fungi pure culture, I have no concern about it. However, in fecal samples, there would be plant residuals containing plant DNA and host DNA (in other words, DNA from eukaryotes). How can you avoid the amplification for LSU from plant and host DNA? Have you tested the potential contamination?
4. The authors used 40 cycles to amplify the targeted region. I am really concerned about this high number of settings. Even if you conduct this amplification only based on the blank regnant, I expect you will also see amplifications. Have you put any positive/negative controls to support that there is no unspecific amplification in your system?
5. As one of the most essential and well-known factors driving gut microbiome, diet has been overlooked in this study. When the authors claim their observation of phyllosymbiosis, it may be merely caused by the differences in diet preference among different species. Could you please add the analyses of diet effect, or discuss this as one of the major limitations of this study?
6. Shotgun metagenome is much more powerful than LSU gene-based amplicon sequencing; why did the author choose amplicon-sequencing instead of shotgun metagenomics?
7. If multiple copies of LSU rRNA gene exist within one fungi species, how could you distinguish them when they have variations? Is it possible that any novel fungi claimed by the authors are just additional copies of existing species or chimeras?
8. Transcriptomic analysis. I did not understand the rationale to conduct this analysis? The expression profile will depend on the growth stage of fungi, media, temperature, and many direct-related environmental factors. They are intron and exon, as well as potential splicing variations for RNA-Seq of fungi genome, which may mislead the analysis. If the authors want to capture the phylogenetic information of fungi, you may want to use genomic sequencing and then assemble them. Therefore, could you explain why you decided to use transcriptomic analysis?

Minor concerns:

1. Inconsistency: in the abstract, the authors mentioned "34" species, while in the supplementary methods, this number was listed as 33. Please double-check.
2. Citation 3-11: these citations are old and outdated if we look the publication dates. Could you please use the most updated references?
3. For the Introduction section, please be more specific and accurate regarding the literature summary. Within my knowledge, gut mycobiome, for example, rumen mycobiome, has been studied pretty well. So please don't over-highlight the knowledge gap or overstate the novelty of the current study.
4. Line 128: could you explain why you decided to target the hypervariable region 2 of the LSU?
5. Fig 1B: logarithmic scale may mislead the readers because it does not tell the possible over-representation of cattle/goats etc., while under-representing many non-domesticated species.
6. Fig 1C: color codes are very similar between different genera, for example, NV7 vs NY46, Other novel genera vs NY50. I suggest using different colors or choosing another strategy to show the data.
7. Fig 1D: the color contrast (between 0 and 229) of the "number of animals" is not obvious enough.
8. Fig 3A-3H: should the statistical analysis results (ab, *, p values) be shown in the figure?
9. Line 238-239: could this be driven by the similar diet among individuals belonging to the same species/family?
10. The Discussion section is too long, and several paragraphs are actually presenting results. The take-home message is not highlighted sufficiently enough.

11. Lines 377-412: only one reference is provided in these paragraphs, meaning most of these words are authors' speculation and/or based on results generated from the current study but without sufficient existing evidence from others, which makes the discussion relatively weak.
12. I feel it is not necessary to repeat the data interpretation in the discussion part (e.g., Line 400-412; Lines 454-460). You can either add this information to the Results section or delete it if it does not provide additional information.
13. The resolution of Fig S4 is not high enough.
14. Please carefully check and correct the misuse of punctuation marks throughout the manuscript.

Reviewer #3 (Remarks to the Author):

Patterns and determinants of the global herbivorous mycobiome

I would like to commend the authors on a worldwide effort and a comprehensive approach to characterize the anaerobic gut fungal mycobiome of mammalian herbivores. AGFs are pivotal in the gut microbiome, However, they are largely unexplored in comparison to other gut microbes. The authors have created a vast data set containing 661 different samples taken from 34 different mammalian species. This data set is unique and possess high potential. However, the manuscript does not seem to have a clear straightforward hypothesis driven line and It is overloaded with exploratory analyses which don't always serve a specific point being conveyed. This makes the reading a bit laborious and unclear. Additionally, the dataset has some limitations which do not seem to be accounted for in the analyses or in the detailed conclusions. As mentioned the data is composed of 661 different animal samples; 538 of these are driven from 4 largely domesticated species (i.e. cow, goat, sheep and horse). The represented non-domesticated samples composed of 97 samples which span 31 different animal species. The large differences in diversity and sample sizes make it somewhat problematic to compare groups such as domesticated/ non domesticated, different species or species hosts without controlling for these discrepancies. Moreover, the manuscript lacks functional knowledge of the gene content and coding capacity of the mycobiome that could have been achieved by various sequencing platforms and could have made this study more robust and important to the field. Overall, though this study has high importance and novelty, it needs a substantial revising in data generation, in depth analysis and writing.

Abstract:

Line 37 : wording is unclear- "beyond current estimate " what are the estimates?

Line 41-42: "phylosymbiosis" is an evolutionary definition. It is not defined by which the composition is determined by the host species or its environment.

Results:

Line 123 & Fig. 1b: gives the impression of high replicability across the samples species which is not the case. Fig. 1b- samples counts should not be in log scale. In addition the text should emphasis the animals with scarce data in addition to the animals with very abundant data.

Line 130: "absolute majority of genus-level diversity was captured". Rarefaction curves in fig S1 indicate that a large portion of the samples did not reach rarefaction.

Line 138: why/ how these 61 samples were chosen for additional se quenching? Please elaborate.

Line: 149: Table S3 is the OTU table, there is no point in referring the reader to it within the results section. It should be under "data availability".

Line 161- 164: Its unclear how the mentioned pattern, is manifested in fog. 2g. In addition fig s5.

Line 177- 184: For Fig. S5, I would expect to see some quantification of the similarity between the trees. or at least plotting the relevant nodes next to each other. This is hard to infer the shared signal as it currently stated. Table S7 is an unintuitive way of comparing between two trees. I would suggest to plot the two trees using the r package "dendextend" to visually emphasize

similarities/ differences.

Line 186-188: please elaborate where do the NST analysis support determinism and where does it indicate stochasticity.

Line 199-200: "Results (Fig. 3i) confirmed a lower overall level of stochasticity in hindgut fermenters, similar to the patterns observed using NST values." please quantify and display statistics to back it up.

Line 219- 222: The ordinations in fig.4 are narrow and one can not conclude from them.

I suggest to reduce them to one or two which answers the question in hand.

Line 235: Please provide titles for the panels in fig. 4c.

Line 243- 247: Please shortly explain what minimal residuals mean.

Line 264- 267: There is a discrepancy between the Animal species analysis and the family analysis. For instance mule, donkey and Zebra have strong associations with *Pinomyces* which does not seem to appear in the family level LIPA analysis. This can be driven by the very high differences in sample sizes between these animals which should be accounted for.

Line 272- 273: This can also mean that different species within the family vary in composition, and that there is no composition that signifies the entire family. Or alternatively that the connection is weaker. This also correspond with the NST analysis that shows less determinism in the ruminants.

Line 276: please tone down. "Corroborate" would be a bit more cautious than "confirmed" since this is a whole different analysis with only 25% of species.

Line 279: the referral to Fig. 6 is absent from this section.

Line 282- 286: the sentence is unclear. What and whose are the preferences in hand? Please rephrase.

Line 285: why were these 20 AGF chosen?

Line 313- 316: what kind of discourse? its not clear. Also please add animal host to figure 6 so the reader will be able to follow the trend described.

Line 320- 321: please tone down. This is correlation and not causation.

Discussion:

Line 339- 342: This statement is true regarding cow, sheep, horse and goat. All are domesticated and comprising 538 out of the 661 samples.

Other animals suffer from a much lower sample size (as much as one sample per animal (e.g. gazelle, mountain goat, takin and more...))

Line 347- 356: Please provide references for these ecological theories.

Line 377- 387: Please provide references.

Line 391-393: Please provide references.

Line 430: there is no evidence that the evolution of the forestomach drove the diversification of AGF. it seems likely since they seem to occur roughly at the same time.

Line 440-442: The sentence should be toned down. It is not shown to have triggered this diversification.

Line 445 -450: A) this belong in the results section. B) these factors also contribute also to the animals which are not mentioned in this specific analysis. This can possibly affect several important analyses such as the LIPA and the PACo. this should be accounted for.

Throughout the manuscript:

Please provide and display relevant statistics within the different figures. It is unintuitive and confusing to search for it in the text.

Captions should be concise and contain only text which is directly relevant for the understanding of the figure.

The ordinations in Fig 4 and Fig S9 show a clear "horse shoe" pattern. Please note that this has to do with specific traits of the data set used to construct these ordinations.

Fig. S7: which PCoA was used? (there are 3 in Fig. 4) In addition, a control is missing (euclidian distance from other centroids) to show its not an artifact of the plot itself.

Fig. S8: Which top 25%? abundance across all animals? In average? Needs to be clarified.

Table S6: the table is missing.

Response to reviewers' comments on Nature Communications NCOMMS-22-49564-T

Reviewer #1 (Remarks to the Author):

This large metabarcoding study investigates the poorly explored communities of anaerobic gut fungi living in association with herbivorous mammals (the herbivore mycobiome). The host sampling scheme is quite large, encompassing herbivores from across the globe (although concentrated in the United States), spanning significant taxonomic breadth, and falling into three major categories (i.e., foregut-fermenting ruminants, foregut-fermenting pseudoruminants, and hindgut fermenters). Noted by the authors, the taxonomic sampling scheme associated with this study includes some never- or rarely sampled herbivore hosts. The authors identify at least 56 novel genera and 4 novel families, dramatically increasing the depth of Neocallimastigomycota sampling. They estimate the impact of stochastic and deterministic effects on driving differences in mycobiome structure, finding that stochastic effects (e.g., drift) play a larger role than previously appreciated and that, interestingly, domestication of herbivores has little apparent affect on mycobiome assemblage. Using a time-calibrated tree based on transcriptome data in conjunction with known diversification times for herbivorous hosts, the authors suggest that coevolution of gut fungi with their hosts has played a more important role than host-filtering in shaping present day gut communities.

Thank you so much for the accurate and well-articulated summary!

This manuscript chronicles a significant amount of work and presents findings that will be interesting to a diverse biological audience, whether they be centered predominantly in mycology, zoology, or elsewhere. The manuscript is well-written (save for some minor typos) and the analyses conducted are appropriate and sufficient toward asserting the conclusions it draws. The authors should be commended for the lengths they have gone to to verify their results, including the use of multiple primer pairs and sequencing platforms as well as the use of dynamic cutoffs for genus level delineations. Given the density of data and breadth of important insights described here, I think this manuscript will find a well-deserved home in Nature Communications. At a finer level, I think there are some points that could be better clarified and some changes that should be made to several figures before that can happen.

Again, thank you so much for the positive assessment.

Major comments and suggestions

1. Given the broad audience of Nature Communications, I think the article could benefit from a more basic and concise description of the different types of herbivore guts delineated in the manuscript (as it is a central concept). I am not an expert in this area and found myself going back to the introduction multiple times to check the meanings of terms, identify which type of gut different animal groups had, etc. I think that this suggestion could be effectively addressed with a figure that diagrams the major anatomical differences between different gut types and places them in an evolutionary perspective (e.g., a small diagrammatic tree of when they evolved and which animal groups have which gut type). I

think this small step to bring readers naive to herbivore gut anatomy (such as myself) will make it easier for the authors to get their exciting conclusions across downstream.

Thank you for the valuable suggestion. In the revised manuscript, a new figure highlighting differences between various gut types was added (Figure 1a in the revised manuscript). As well, a description of the differences between gut types is provided in the introduction (L63-73 in the revised manuscript).

2. Throughout the manuscript, the finding that a portion of novel genera were only ever found at low abundances (5/56 \leq 0.1% and 15/56 \leq 1%) is used to make the point that these taxa are perpetually rare. The authors make this assertion stronger by their having factored out the impact of primer bias (by using multiple primer pairs), sequencing technology (by using Illumina and PacBio chemistry), and sequencing libraries are sufficient depth. While I commend the authors for their attention to conducting meaningful controls and replication to this end, I think that the impact of (especially) primer bias, cannot be completely ruled out and should be mentioned more often as a possible driver of results. PCR always introduces bias and these primer pairs were necessarily designed agnostic to the many new taxa that they discover in this study. It is not surprising then that many of these novel genera are often present at such low abundances. Can the authors please either (i) clarify how they have sufficiently mitigated primer bias to the point where it can be ruled out (ii) add text when mentioning this assertion of rarity that clarifies the potential impact of primer bias (e.g., L164-167, L347-348, etc.). As a minor point in the same vein, I would caution the authors against fully synonymizing read abundance with true abundance.

Thank you, we appreciate the reviewer's assessment. As the reviewer states, efforts have been made to mitigate the impact of primer errors. In the revised manuscript, we followed the reviewer's second suggestion in acknowledging the possible potential impact of primer bias, and the impossibility of completely ruling it out for some genera (L414-419 in the revised manuscript).

3. I think the coloring and organization of some of the figures could be improved toward concise communication of the conclusions.

• Figure 1C: There are color pairs in the rightmost legend (accompanying the stacked bar plot) that are going to be imperceptible to readers – there are just too many. I would suggest either (i) summing genera below some percentage into a single category or (ii) if you must keep each genus separate, order the legend such that it matches the stacked bar from high to low proportion. Also, the text alignment overlaid the stacked bar needs to be tightened up – on my PDF some are overlapping or misaligned with each other.

Thank you, we agree. The same issue was also raised by reviewer 2. To improve figure 1C, we followed the reviewer's second advice in which the order of the legend matches the stacked bar from high to low. The figure is also now larger in size.

• Figure 1D: On the right bar charts, some background shading (e.g., alternative grey and white) between different fungal taxa (in addition to the bar coloration) would help the

reader delineate which bars correspond to which taxon. It's not difficult when the bar coloration is visible, but in many cases it is not (e.g. *Joblinomyces* – *Buwchfawrowmyces*). Thank you, we experimented with extra-shading, but this did not really improve the clarity of the figure for genera with low abundance.

• **Figure 3A-H: Significant comparisons should be marked here and p-values could also be shown on the plots.**

Thank you, we agree. Significance asterisks have been added to the revised versions of Figure 3.

• **Figure 3I: Could use a y-axis label, despite it somewhat obviously being percentage.**

Yes, thank you, y-axis label (percentage contribution to AGF community assembly) has been added to the revised figure.

• **Figure 4A: Could the points on the NMDS/PCoA plots be smaller? At present, the degree of overlap makes it is hard to visually see when points more or less clustered along the axes.**

Thank you. We agree. We decreased the size of the points on all plots in the revised manuscript. As well, the RDA and NMDS plots were relegated to the supplementary document, as per additional suggestions by reviewer 3 (see below).

• **Figure 5D: It looks like some color legends are missing here.**

Thank you. Yes, we apologize. We added the needed legends to Figure 5D to describe the AGF genera coloring and the host species and family coloring.

Minor comments and suggestions

L171. Unmatched parentheses

Thank you, parenthesis removed.

L171. 'formed additional' should be 'formed an additional'

This was changed to 4 additional well-supported family-level clusters to avoid confusion.

L171-174. The authors might consider using the same sentence structure here (new families) as in the sentence above (accepted families) instead of the long piece of text in parentheses.

Thank you, agreed. The same sentence structure is used now in both sentences.

L370. I'd rewrite to: 'where the roles of phylogenetically distinct taxa are ecologically interchangeable.'

Thank you, agreed. The sentence was changed as suggested.

L374. 'community' should be 'communities'.

Yes, changed to "communities".

L374. 'readily surmounted' is strong language – are the described processes (e.g. post-birth grooming) that effective are circumventing this barrier?

Thank you. It is certainly hard to accurately quantify how effective post-birth grooming, or other early contact between mothers and offspring is in overcoming dispersal limitation. We agree that readily surmounted is a strong way to state a possible explanation to an observed pattern. In the revised manuscript, we changed “is readily” to “could plausibly be”.

L388. ‘factor’ should be ‘factors’.

Correct. Changed in the revised manuscript.

L455. ‘explaining’ should be ‘explained’

Correct. Changed in the revised manuscript.

L454-457. incomplete sentence as currently written

Thank you, the sentence was reworded to a complete sentence.

Supp-L452. I’m not sure if this has to do with how I downloaded my particular version of this file, but Figure S4 is of poor quality (pixelated).

We apologize, we are not sure why this occurred during file conversion. The updated version in the revised manuscript is of a higher resolution

Reviewer #2 (Remarks to the Author):

The authors profiled the mycobiome of 661 fecal samples (sufficient sample size) from multiple herbivorous species, which further expanded the research scope of anaerobic gut fungi and increased our understanding of the diversity of gut mycobiome. The biological question raised here is essential. The bioinformatic analyses are logical, and the data interpretations are sufficient enough. However, I still have several major and minor concerns about the current study that need to be further clarified and addressed.

Major concerns:

1. For ruminal species included in this study, I don’t think the fecal mycobiome could represent the actual function of the gut mycobiome inhabiting these hosts; most of the fungi play functions in the rumen instead of the hindgut. In this case, why did the authors choose fecal samples?

Thank you very much. The reviewer raises an important point. We chose fecal over rumen samples for multiple reasons:

1. The current effort aims for a wide characterization of the AGF community in ruminants, pseudoruminants, and nonruminant hindgut fermenters. Pseudoruminants lack a true rumen and nonruminant hindgut animals completely lack pregastric chambers. For comparative reasons, it would not be appropriate to compare samples taken from different GIT locations. Fecal samples are the common specimen that could hence be used for comparative purposes.
2. Fecal sampling also circumvents logistical and ethical issues associated with sampling animals’ internal structures, e.g. euthanization, fistulation, or gastric tube insertion to obtain rumen and diverticula samples from ruminants and pseudoruminants. Also, there are

insurmountable logistical hurdles associated with obtaining such samples from wild animals or from precious zoo possessions.

3. Fecal samples are homogenous, which render them more amenable to comparison, and their sampling process could easily be standardized across laboratories. In comparison, rumen samples are extremely heterogenous, with liquid and solid components. The microbial community could greatly differ between particle-associated and liquid-associated populations (e.g. Pubmed ID: 35511785). This can certainly complicate sampling, reproducibility, and comparative analysis.

For these reasons, we felt that fecal samples are more appropriate. In the revised manuscript, we added wording in the methods section to justify our choice (L539-550 in the revised manuscript).

2. DNeasy Plant Pro Kit (Qiagen®, Germantown, Maryland) was applied to extract fungi DNA from fecal samples. Is this a validated protocol for isolating fungi DNA? Are there any previous studies testing the efficiency and accuracy of this kit in terms of fungi DNA isolation?

Thank you. The community of AGF scientists have worked and agreed on using this kit as the preferred method for DNA extraction. The efficiency has been validated in a recent publication (PMID: 36144352). We added this information to the revised manuscript (L553-554 in the revised manuscript).

3. The authors chose LSU rRNA gene as the targeted region. If this is only for fungi pure culture, I have no concern about it. However, in fecal samples, there would be plant residuals containing plant DNA and host DNA (in other words, DNA from eukaryotes). How can you avoid the amplification for LSU from plant and host DNA? Have you tested the potential contamination?

Thank you, both the Illumina (D2 LSU), and SMRT (D1/D2 LSU) primers used in this study have been developed and thoroughly validated for coverage and specificity to the Neocallimastigomycota in prior publications from the AGF community (PMID: 36144352 and 29423636). In addition to in-silico work confirming their specificity, a recent manuscript (PMID: 36144352) has done an excellent job confirming the specificity of these primers by testing them in a wide range of environments. In that study, AGF amplicons were successfully obtained and sequenced from fecal samples of herbivores. However, the primers failed to detect AGF in environments where they are not present (e.g., animal feces of carnivores, and soils). Many of the authors on that manuscript are coauthors on this manuscript as well.

Second, our bioinformatic pipeline failed to identify LSU sequences belonging to other fungi, plants, or hosts in our analysis, further demonstrating the specificity of these primers. Our quality control identified and removed some low-quality sequences, but none of which were non-target LSU sequences.

For these reasons, we are confident that no potential contamination or sequences that are not affiliated with the Neocallimastigomycota were included in our datasets.

4. The authors used 40 cycles to amplify the targeted region. I am really concerned about this high number of settings. Even if you conduct this amplification only based on the blank

regnant, I expect you will also see amplifications. Have you put any positive/negative controls to support that there is no unspecific amplification in your system?

Thank you so much. Yes, given the low DNA content of fungi, compared to bacteria and archaea, this is a valid concern. We were extremely careful to avoid contamination and non-specific amplification. To this end, negative (reagents only) controls were always run with every batch of PCR amplification reactions. Negative controls yielded no amplification bands in the absolute majority of PCR batches. In rare cases, very faint bands were observed in the negative control. In such case, all amplicons were not further purified and processed for sequencing, and the extracted DNA from these samples were discarded. After a thorough cleanup and autoclaving of all pipettes and tips, DNA was re-extracted from these samples and PCR amplification was reconducted. Under no circumstances were amplicons sequenced in runs where a faint band was observed with the negative reagent-only control. This information was added to the revised manuscript (L564-565 in the revised manuscript).

5. As one of the most essential and well-known factors driving gut microbiome, diet has been overlooked in this study. When the authors claim their observation of phyllosymbiosis, it may be merely caused by the differences in diet preference among different species. Could you please add the analyses of diet effect, or discuss this as one of the major limitations of this study?

Thank you, this is certainly true in a general sense. However, we would like to note that in this study, a more nuanced view of the putative role of diet in shaping AGF community should be considered. Unlike prior work that examined the effect of diet on bacterial and archaeal diversity between herbivores, omnivores, and carnivores (e.g. PMID: 31685619); the magnitude of diet-induced differences, if any, in AGF diversity might not be as drastic. This is due to the broad overall similarity in diet and similar metabolic function (breaking down plant biomass) undertaken by all known AGF genera.

Further, we feel that the assessment of the impact of diet using the current dataset is not ideal, given the fact that many animal species in the dataset appear to have highly similar diet (especially domesticated cattle, horses, goats, and sheep, as well as captive animals housed in the same zoo). As well, exact documentation of diet in wild herbivores over a long time span is infeasible.

We acknowledge, as the reviewer states, that this is a limitation of this study, and argue that a more targeted effort in which diet is purposefully manipulated in a similar cohort(s) of one animal species and monitored over a prolonged time frame is needed to address such issue. In the revised manuscript, we added wording to that effect in the discussion (L316-323 in the revised manuscript).

6. Shotgun metagenome is much more powerful than LSU gene-based amplicon sequencing; why did the author choose amplicon-sequencing instead of shotgun metagenomics?

Thank you, in principle, we completely agree that metagenomics is much more powerful when compared to amplicon-based surveys, as it provides additional functional insights into the community. However, when considering conducting shotgun metagenomics on AGF from fecal samples, three huge hurdles exist:

First, the high cell numbers of bacteria and archaea compared to fungi in the examined samples render recovering AGF sequences from the metagenomic sequencing output virtually impossible. Prior studies reported bacterial count in fecal samples in the 10^{10} - 10^{11} range (e.g., PMID 18652685), while fungal count rarely exceeds 10^4 thallus forming units per gram of fecal samples (PMID 16348156). Further, a sizable fraction of anaerobic gut exists as zoospores in the GIT, rather than large vegetative structures. Even when existing as vegetative structures, such hyphae have much lower DNA/ gram biomass compared to bacterial cells.

Second, the scarcity of AGF reference genomes renders assigning AGF contigs (if encountered) to their appropriate genus highly tentative. Currently, genomes from representatives of only 5 out of 20 cultured genera (*Neocallimastix*, *Anaeromyces*, *Piromyces*, *Pecoromyces*, and *Caecomyces*) are available. Further, unlike bacteria where single cell genomics and genome assembly from metagenomes approaches enabled generation of reference genomes from uncultured taxa, no similar resource or reference genomes from anaerobic gut fungi are currently available. In other words, the lack of reference genomes essentially precludes effective phylogenetic anchoring of AGF sequences (if obtained) in any proposed metagenomic efforts.

Third, AGF genomes sequenced so far have a very low coding density and extremely low GC content especially in intergenic regions. In such regions long stretches of >100 bp Adenines (A) or Thymines (T) are found. This architecture renders assembling AGF genomes in pure cultures and metagenomic datasets highly challenging.

Indeed, given such difficulties, it is not surprising that, so far, only one single prior study has successfully recovered AGF contigs using metagenomics (PMID: 23709508). However, this was only achieved in targeted enrichments, where antibiotic addition promoted AGF growth and inhibited bacterial growth. This enrichment approach yielded an AGF monoculture, with all AGF sequences recovered putatively assigned to a single genus (*Neocallimastix*). As such, this approach essentially obliterates AGF diversity and negates the value of using metagenomics for AGF diversity assessments.

We have added such arguments to the discussion section of the revised manuscript (L513-528 in the revised manuscript).

Beyond these theoretical arguments, we also undertook multiple additional efforts to evaluate the use of metagenomics to obtain detailed insights into AGF diversity and community structure in mammalian feces.

1. To further confirm that AGF DNA is extremely sparse in fecal samples, we quantified AGF in 10 cattle, 10 goat, 10 sheep, and 10 horse samples using qPCR, and compared the AGF numbers to the total bacterial numbers in the same samples. Our results, now added to the revised manuscript, confirmed that bacteria outnumber AGF by at a factor of 1:250,000 rRNA gene copy number between AGF fungi and bacteria. We present this information in Figure S13 in the revised manuscript. As well, the methods sections in the main and supp. document, as well as the results and discussion sections in the main document were amended to describe such effort (L386-390, 518-519, and 701-709 in the main document, and L418-442 in the supp. document)

2. To further demonstrate that metagenomic efforts could not detect AGF in feces, we conducted a literature search on metagenomic studies of fecal samples from herbivores in 29 studies, each of which contains multiple fecal metagenomic samples. The information is presented **in Table S16** in the revised supplementary document. In 28 out of 29 studies, no AGF affiliated reads, contigs, or bins were identified. Only a single study, that generated 6.5 Tb of data, identified 13 bins of AGF out of the total of 28,543 metagenomic bins assembled (Table **S16 Ref 52 in the supplementary document**). These bins were small, and collectively, represented less than 1% of a reference AGF genome. Further, due to the lack of adequate reference genomes, the manuscript's assignment to the AGF genera *Piromyces* and *Anaeromyces* were highly tentative. Finally, due to the high density of non-coding regions, the authors were not able to provide any meaningful discussion regarding the identity and function of AGF genes in these contigs. Collectively, the small amount, uncertain affiliation, and lack of adequate gene content renders using this information for diversity assessment and documentation unfeasible.

3. Nevertheless, we reasoned that there might be a remote possibility that failure to detect fungal DNA in prior studies might be a reflection of the use of appropriate DNA extraction methods or the use of outdated bioinformatic approaches. Therefore, **we decided, nevertheless, to fulfill the Editor and reviewer 2 request and conduct our own metagenomic sequencing on a subset of the data.** To this end, we conducted deep shotgun metagenomic sequencing on nine fecal samples. Within the entire 611 M reads obtained, only 1.56 M, and 48K reads were assigned to eukaryotic origins (using both Kaiju, and GOTTCHA, respectively) (**Table S13 and S14**), and only 38K of these were assigned to AGF, all of which were present in 1/9 sample, while 8 out of the 9 samples contained no AGF reads (**Figure S12**). Further, to account for the possibility that identification of AGF at the illumina reads level might be hampered by the short read length and the low coding density in AGF genomes, reads were assembled into contigs. EukRep, a *k*-mer-based strategy for eukaryotic sequence identification in metagenomic datasets (PMID:29496730), was further used to identify contigs potentially belonging to eukaryotic organisms. Our analysis (**Figure S15**) failed to identify a single AGF contig in all datasets.

The metagenomic effort conducted is presented in Figures S12 and S15, and Tables S13 and S14. In addition, text describing such efforts are presented in the main document in L268-385, L513-528, and L692-700, and in the supplementary document L397-417)

Collectively, we feel that we provide irrefutable experimental evidence (metagenomic sequencing and qPCR data) to demonstrate the infeasibility of using shotgun metagenomic sequencing for diversity assessment and documentation for AGF in herbivorous feces. We also provide clear explanation of factors underpinning such infeasibility.

7. If multiple copies of LSU rRNA gene exist within one fungi species, how could you distinguish them when they have variations? Is it possible that any novel fungi claimed by the authors are just additional copies of existing species or chimeras?

Thank you, this specific issue has been intensively investigated by many scientists exploring AGF diversity (all of which are coauthors on this paper) in the last few years. Prior work (PMIDs: 32656919, 36144352) has convincingly demonstrated that, while multiple LSU copies are indeed present in any AGF genome, the level of LSU sequence divergence within a single

genome is extremely low (0.13% to 1.84%). These thresholds are certainly below those used here for assigning taxa to genera (5%). Indeed, such characteristic, and uniform length was the main reason the AGF community has transitioned from using ITS1, the most commonly used phylomarker for fungi, to LSU for diversity analysis.

8. Transcriptomic analysis. I did not understand the rationale to conduct this analysis? The expression profile will depend on the growth stage of fungi, media, temperature, and many direct-related environmental factors. They are intron and exon, as well as potential splicing variations for RNA-Seq of fungi genome, which may mislead the analysis. If the authors want to capture the phylogenetic information of fungi, you may want to use genomic sequencing and then assemble them. Therefore, could you explain why you decided to use transcriptomic analysis?

Thank you. Transcriptomic analysis here is done with the goal of obtaining the gene content of each strain that we have, rather than assessing expression patterns of a single or few species. The use of RNA sequencing for obtaining gene content is a common practice in fungal research (PMID:35900180), including the AGF (PMID: 31126947, 36827202, 30061875). This is particularly driven in the AGF by the difficulties associated with obtaining genomes due to their extremely AT rich genomes (especially in the intergenic regions), large noncoding intergenic regions, and proliferation of microsatellite repeats (PMID: 23709508). Transcriptomics has the added value of generating only coding sequences with *no introns*, compared to genomes. As such, while the value of obtaining fungal genomes is generally well taken, we feel that the transcriptomic dataset provides the necessary information for conducting the phylogenomic analysis presented in this study. We added a few sentences in the methods section of the revised manuscript to explain our rationale (L655-658 in the revised manuscript).

Minor concerns:

1. Inconsistency: in the abstract, the authors mentioned “34” species, while in the supplementary methods, this number was listed as 33. Please double-check.

We apologize. Thirty-four is the correct number. This was fixed in the supplementary document.

2. Citation 3-11: these citations are old and outdated if we look the publication dates. Could you please use the most updated references?

Thank you. For references 3-8: these are general references regarding the structure of the GIT for herbivores, and no new information or findings on the subject has been reported since the publication of these references.

Regarding references 9-11, these are the original papers that described the discovery of the AGF in the herbivorous gut (in the 1970s), and hence could not be updated.

3. For the Introduction section, please be more specific and accurate regarding the literature summary. Within my knowledge, gut mycobiome, for example, rumen mycobiome, has been studied pretty well. So please don't over-highlight the knowledge gap or overstate the novelty of the current study.

I assume that the reviewer refers to the section in L65-82. We note that in the original manuscript, we acknowledged that several studies have looked at the gut mycobiome using high throughput sequencing (most of which conducted by laboratories involved in this effort). We

even provided a table (Table S1) with a list of these studies. As obvious by the rather short list, there are not many. We did a thorough literature check, but of course would be happy to add more studies, if pointed out by the reviewer.

As such, we stand by our assessment that a significant gap of knowledge exists, that is addressed by this study. Nevertheless, we reworded some parts of the text in the revised manuscript to be more specific and highlight that while prior studies exist, they are by far fewer than studies on the bacterial and archaeal component of the GIT (L82-85 in the revised manuscript).

4. Line 128: could you explain why you decided to target the hypervariable region 2 of the LSU?

The choice of the primers is a culmination of multiple efforts by the AGF community for a suitable marker for diversity survey. Earlier studies have provided multiple arguments for the merit of using LSU, rather than ITS1 for AGF diversity assessment (see PMID:31681229, 32656919). A recent diversity survey utilized a 700 bp regions (D1/D2 region for diversity assessment) (PMID:32656919). However, the length of the amplicon necessitates using Sanger or SMRT sequencing, a more complicated endeavor than Illumina. Recently, coauthors Young et al (PMID:36144352) have shown that a shorter region encompassing the D2 only provides similar coverage, resolution, and specificity. Details on correlation of data generated using D1/D2 LSU versus D2 LSU is provided in great detail in the supplementary document in the original manuscript (now L122-135 in the revised supp. document).

5. Fig 1B: logarithmic scale may mislead the readers because it does not tell the possible over-representation of cattle/goats etc., while under-representing many non-domesticated species.

Yes, thank you, that is true. It is not our intention to mislead, but rather to present the data in a readable way. The issue has also been raised by reviewer 3. This figure has now been removed and information on the number of samples per animal species, and the domestication status is now added to Figure 1d.

6. Fig 1C: color codes are very similar between different genera, for example, NV7 vs NY46, Other novel genera vs NY50. I suggest using different colors or choosing another strategy to show the data.

Thank you, yes this was also highlighted by reviewer 1, we have reworked the figure for clarity, as described above.

7. Fig 1D: the color contrast (between 0 and 229) of the “number of animals” is not obvious enough.

Thank you, that is true. To address that issue, we have now added the actual numbers to the updated Figure 1d.

8. Fig 3A-3H: should the statistical analysis results (ab, *, p values) be shown in the figure?

Thanks, we agree. This has been added to the figure. This has also been raised by reviewer 1.

9. Line 238-239: could this be driven by the similar diet among individuals belonging to the same species/family?

Yes, this is partly possible. As we state above, since all these animals are herbivores, and since most subjects within a species eat the same food, disentangling importance of diet from species is extremely hard as described above.

10. The Discussion section is too long, and several paragraphs are actually presenting results. The take-home message is not highlighted sufficiently enough.

Thank you, we agree. We removed some parts of the discussion, especially sections that restated information from the results. Specifically, L402-411, and L448-463 from the original manuscript were removed from the discussion.

11. Lines 377-412: only one reference is provided in these paragraphs, meaning most of these words are authors' speculation and/or based on results generated from the current study but without sufficient existing evidence from others, which makes the discussion relatively weak.

Thank you, multiple references were added to this section in the revised manuscript (L452-483 in the revised manuscript).

12. I feel it is not necessary to repeat the data interpretation in the discussion part (e.g., Line 400-412; Lines 454-460). You can either add this information to the Results section or delete it if it does not provide additional information.

Thank you, as described above, the first instance (L400-412) was removed in our effort to shorten the discussion. The second part was moved to the results. A similar request to move this section to the results has also been made by reviewer 3.

13. The resolution of Fig S4 is not high enough.

Yes, thank you, this has been raised by reviewer 1 as well. Fixed in the revised submission.

14. Please carefully check and correct the misuse of punctuation marks throughout the manuscript.

Thank you, we went through the manuscript and corrected multiple instances of improper punctuation.

Reviewer #3 (Remarks to the Author):

Patterns and determinants of the global herbivorous mycobiome

I would like to commend the authors on a worldwide effort and a comprehensive approach to characterize the anaerobic gut fungal mycobiome of mammalian herbivores. AGFs are pivotal in the gut microbiome, However, they are largely unexplored in comparison to other gut microbes. The authors have created a vast data set containing 661 different samples taken from 34 different mammalian species. This data set is unique and possess high potential.

Thank you very much for the kind words.

However, the manuscript does not seem to have a clear straightforward hypothesis driven line and It is overloaded with exploratory analyses which don't always serve a specific

point being conveyed. This makes the reading a bit laborious and unclear. Additionally, the dataset has some limitations which do not seem to be accounted for in the analyses or in the detailed conclusions. As mentioned the data is composed of 661 different animal samples; 538 of these are driven from 4 largely domesticated species (i.e. cow, goat, sheep and horse). The represented non-domesticated samples composed of 97 samples which span 31 different animal species. The large differences in diversity and sample sizes make it somewhat problematic to compare groups such as domesticated/ non domesticated, different species or species hosts without controlling for these discrepancies. Moreover, the manuscript lacks functional knowledge of the gene content and coding capacity of the mycobiome that could have been achieved by various sequencing platforms and could have made this study more robust and important to the field.

Overall, though this study has high importance and novelty, it needs a substantial revising in data generation, in depth analysis and writing.

Thank you for the constructive criticism. We hope our additional efforts, highlighted below, and added to the revised manuscript addresses the reviewer's comments satisfactorily.

Abstract:

Line 37: wording is unclear- “beyond current estimate “ what are the estimates?

The current estimates are 31 genera and candidate genera, as stated in the results section in the original manuscript. These numbers have been added to the abstract in the revised manuscript.

Line 41-42: “phylosymbiosis” is an evolutionary definition. It is not defined by which the composition is determined by the host species or its environment.

Correct, thank you. Host identity is meant here as host phylogeny and evolutionary relationship between various hosts, rather than specific traits in the host and its environment. We have changed the sentence to “host phylogenetic affiliation (animal species, family, and associated gut type..” for accuracy.

Results:

Line 123 & Fig. 1b: gives the impression of high replicability across the samples species which is not the case. Fig. 1b- samples counts should not be in log scale.

Thank you. The same issue was raised by other reviewers. As stated above, it was not our intention to deceive the reader to falsely implying equal number of samples for all animals. Rather, we meant to make finding the actual number of samples per animal easier for both high and low replicate animals. This figure has now been removed and tracks with numbers and domestication status were added to Figure 1d.

In addition the text should emphasis the animals with scarce data in addition to the animals with very abundant data.

Thank you, we agree. To highlight the scarcity of some samples and abundance of others, we added the number of samples after every animal. The earlier version only gave the numbers of samples for well sampled animals. The revised sentence (**L131-138 in revised manuscript**)

However, we hope the reviewer recognizes the difficulties of obtaining hundreds of samples from animals that could be endangered, elusive, and/or with few subjects available at zoos and the wild, e.g., manatees, rhinoceros, chamois, ibex, lechwe, markhor, okapi, takin, capybara,

Patagonian mara, markhor, and takin. The effort associated with securing samples from such a diverse number of herbivores was not trivial, and involved an untold number of communications, intense negotiations, and considerable cost. As such, we feel that it is logical and expected to have large number of replicates, when possible (e.g. for domesticated animals), and much fewer for others where sample acquisition is much harder.

Line 130: “absolute majority of genus-level diversity was captured”. Rarefaction curves in fig S1 indicate that a large portion of the samples did not reach rarefaction.

We agree. Thank you. Yes, absolute majority should not be used here. We apologize. We changed that sentence to “Rarefaction curve (Fig. S1) and coverage estimates (Table S3) demonstrated that the majority of genus-level diversity (>90% based on Good’s coverage) was captured in 97.7% of samples.” (L142-144 in the revised manuscript).

Line 138: why/ how these 61 samples were chosen for additional sequencing? Please elaborate.

These 61 samples simply represented the first batch of samples that were available and ready for DNA extraction and sequencing during the project. In this early stage, we sought to examine how primer pair or sequencing technology (Illumina) employed would affect the AGF community structure. To this end, we sequenced this first batch (61 samples) using both approaches and compared the results. Once we demonstrated that a highly similar microbial community was observed using both approaches, we proceeded to use the more straightforward and economic Illumina approach. Such approach was useful not only for confirming the similarity in the community obtained by the two approaches but was also useful for the simultaneous recovery of full-length sequence representatives (~700 bp covering the D1/D2 regions) of novel genera obtained.

To clarify this issue, we slightly changed the wording from “ a subset of...” to “the first batch of available samples” (L150-154 in the revised manuscript).

Line: 149: Table S3 is the OTU table, there is no point in referring the reader to it within the results section. It should be under “data availability”.

Thank you. We agree that referring to the OTU table here is not needed and we removed the reference to it in this particular spot. However, we still feel that it should be referenced in the results section and should be kept as a supplemental table.

Line 161- 164: Its unclear how the mentioned pattern, is manifested in fig. 2g. In addition fig s5.

Thank you, yes, Figure 2G does not clearly demonstrate that. We apologize. We removed reference to it here. We think the reviewer is referring to Table S5, not Figure S5 here, and yes, it does not clearly show that as well. Here, Table S3 would be the best place to find this information, and hence we referenced it accordingly.

Line 177- 184: For Fig. S5, I would expect to see some quantification of the similarity between the trees. or at least plotting the relevant nodes next to each other. This is hard to infer the shared signal as it currently stated. Table S7 is an unintuitive way of comparing

between two trees. I would suggest to plot the two trees using the r package “dendextend” to visually emphasize similarities/ differences.

Thank you. In the revised manuscript, we plotted the two trees face to face as proposed with lines extending between the same genus in each tree. **The results are now in the new Fig. S6.** We thank the reviewer for this valuable suggestion.

Line 186-188: please elaborate where do the NST analysis support determinism and where does it indicate stochasticity.

Thank you, the explanation is provided in the supplementary methods section, but we agree that a quick explanation is needed here.

In the revised manuscript, we added additional wording for clarification **(L202-204 in the revised manuscript).**

Line 199-200: “Results (Fig. 3i) confirmed a lower overall level of stochasticity in hindgut fermenters, similar to the patterns observed using NST values.” please quantify and display statistics to back it up.

Thank you, we added such numbers in the revised manuscript. The significance asterisks are now showing on the figure, as also suggested by reviewers 1 and 2.

Line 219- 222: The ordinations in fig.4 are narrow and one can not conclude from them. I suggest to reduce them to one or two which answers the question in hand.

Thank you, this is an issue that has been intensely debated in drafts circulating between the co-authors prior to submission. In accordance with the reviewer’s request, we only kept the ordination based on weighted unifracs in the main figure and moved all other ordination plots to the supplementary material (now Fig S8 in the revised manuscript).

Line 235: Please provide titles for the panels in fig. 4c.

Thank you, done.

Line 243- 247: Please shortly explain what minimal residuals mean.

Thank you, the sentence refers to PACo Procrustes residuals of the sum of squared differences within different animal species. It clarifies that the residuals from Paco analysis were very small. We rewrote this sentence for clarity in the revised manuscript **(L260-265 in the revised manuscript).**

Line 264- 267: There is a discrepancy between the Animal species analysis and the family analysis. For instance mule, donkey and Zebra have strong associations with Piromyces which does not seem to appear in the family level LIPA analysis. This can be driven by the very high differences in sample sizes between these animals which should be accounted for.

Thank you for the valuable comment. The heatmap of LIPA values is an average number of all individual Lipa values of an animal species (left) or an animal family (right) with the corresponding AGF genus. Family Equidae is represented by 5 animal species, with samples from horses representing the majority (n=138). The reviewer is correct about variation in sample size between different hosts, and this was the main reason driving our decision to show the results at both the animal species level as well as the animal family level. We are not sure how to account for the differences in animal species numbers other than to show the results the way we

are presenting them. We added the following text to the figure legend “Note that because average values are shown here, and due to the variation in the number of individuals belonging to each of the animal species, LIPA associations identified with animal species might not always be reflected at the family level (for example: AL3, and *Piromyces* are clearly associated with mules, and donkeys, respectively, but this association was not strong at the Equidae family level due to the small number of mules (n=4) and donkeys (n=5) studied compared to the total number of Equidae animals (n=152))”.

Line 272- 273: This can also mean that different species within the family vary in composition, and that there is no composition that signifies the entire family. Or alternatively that the connection is weaker. This also correspond with the NST analysis that shows less determinism in the ruminants.

The reviewer has a good point. Yes, this could mean that different animals have strong signatures but do not share such signatures with other animals in the same family. This was certainly observed with hindgut fermenters (see our comment above). However, a similar trend was not observed with individual ruminant animals (as apparent from the lack of green (strong) LIPA associations in figure 5b), except for these few that were not deeply sampled (e.g. yak (n=4), oryx (n=4), buffalo (n=11), bison (n=7), miniature Zebu (n=3), elk (n=2)). Accordingly, such strong association in shallow-sampled animals is possibly not animal species-specific, but rather individual animal effect. As well, such strong association were also not translated to the family level given the high number of replicates within families (e.g. Bovidae family with 437 individuals sampled).

Line 276: please tone down. “Corroborate” would be a bit more cautious than “confirmed” since this is a whole different analysis with only 25% of species.

Thank you, changed to “corroborate” in the revised manuscript.

Line 279: the referral to Fig. 6 is absent from this section.

Reference to the figure was at line 294 in the original manuscript (L339 in the revised manuscript).

Line 282- 286: the sentence is unclear. What and whose are the preferences in hand? Please rephrase.

Thank you, the choice of the word “preferences” here was a mistake that made the sentence unclear. We apologize. We changed it to “patterns” which is what we meant.

Line 285: why were these 20 AGF chosen?

These strains, simply put, are the ones that are available in the laboratories of the authors. These strains represent 13/20 genera present, and together with existing ones, cover 14/20 AGF genera. To our knowledge, the remaining six genera not included in the analysis have no live representatives. Given the small number of researchers in the area (essentially >80% are coauthors in this study), we feel that this is the most possible dataset that could be included for sequencing. We should note that AGF maintenance is extremely hard, and currently there is no official culture collection for AGF strains.

Line 313- 316: what kind of discourse? its not clear. Also please add animal host to figure 6 so the reader will be able to follow the trend described.

Thank you. We agree. The sentence as written was unclear. The word discourse here is not a wise choice. We have reworded the sentence to read “While these results suggest the central role played by co-evolutionary phylosymbiosis in shaping AGF community, timing estimates for a few AGF genera did not correspond to the evolutionary history of their preferred animal hosts. Such pattern suggests a time-agnostic post-evolutionary environmental filtering process”. **L357-361 in the revised manuscript.**

Line 320- 321: please tone down. This is correlation and not causation.

Thank you. “argue for a major role” was changed to “suggest”.

Discussion:

Line 339- 342: This statement is true regarding cow, sheep, horse and goat. All are domesticated and comprising 538 out of the 661 samples.

Other animals suffer from a much lower sample size (as much as one sample per animal (e.g. gazelle, mountain goat, takin and more...))

Ture. We changed the wording to “We examined a large number of replicates for several domesticated animal species” **(L407-409 in the revised manuscript)**. The same issue has also been raised by reviewer 2.

Line 347- 356: Please provide references for these ecological theories.

Thank you, we have now provided the references **(Ref 64-70)**, and also added some additional wording for clarity, e.g. as suggested previously in assessments of bacterial rare biospheres **(L420-434 in the revised manuscript)**

Line 377- 387: Please provide references and Line 391-393: Please provide references.

This also has been raised by reviewer 2. We provide references in the revised manuscript **(L452-483 in the revised manuscript)**.

Line 430: there is no evidence that the evolution of the forestomach drove the diversification of AGF. it seems likely since they seem to occur roughly at the same time.

Thank you, changed to “was associated with a parallel”.

Line 440-442: The sentence should be toned down. It is not shown to have triggered this diversification.

Thank you, triggered changed to “putatively facilitated”.

Line 445 -450: A) this belong in the results section.

Thank you, we agree. We moved this section to the results. This has also been suggested by reviewer 2. **Now at L302-315 in the revised manuscript.**

B) these factors also contribute also to the animals which are not mentioned in this specific analysis. This can possibly affect several important analyses such as the LIPA and the PACo. this should be accounted for.

Yes, these factors could also contribute to animals not mentioned in the specific analysis. In principle, any additional data may or may not affect the output of the ordination and community structure work. However, this is not an excuse not to do it nor should it be considered a drawback of such analysis. As we clearly state, the high level of replication required for such an analysis is achieved in only a few animal species. Again, we hope the reviewer appreciate the enormity of the effort involved and difficulty with sampling.

Throughout the manuscript.

Please provide and display relevant statistics within the different figures. It is unintuitive and confusing to search for it in the text.

Thank you, this has been added as request (Figure 3 in the revised manuscript).

Captions should be concise and contain only text which is directly relevant for the understanding of the figure.

Thank you, we shortened them significantly in the revised manuscript as could be seen in the “compare documents” version provided with the revised submission. Briefly, the following was removed:

Figure 1. Removed referring to figure 1b (now merged with 1d in the revised manuscript). Also removed the sentence “ the relative abundances of the novel genera delineated in this study are shown as a stacked column on the right”

Figure 2. Removed “The distribution of the percentage of novel genera is shown as box and whisker plots”

Figure 4: Significantly shortened and modified the figure legend to account for the changes implemented in the figure (moving multiple graphs to the supplementary document).

Figure 5:

Removed “To test the robustness of the phylogenetic signal of association between host phylogeny and the AGF community, PACo analysis was repeated 100 times while subsampling one individual per host genus. The box and whisker plots show...”.

Also removed “The host animal tree and host family tree on top were downloaded from timetree.org. Animals are color coded by their respective family and colors follow the same scheme as in Fig. 1d.”

The ordinations in Fig 4 and Fig S9 show a clear “horse shoe” pattern. Please note that this has to do with specific traits of the data set used to compost these ordinations.

We agree with the reviewer that horseshoe in ordination plots could be due to specific traits within the data. Here, this is most probably caused by different number of sequences between samples.

Since this pattern was mostly observed with plots constructed using Bray-Curtis values, and to guard against solely basing our conclusion on plots demonstrating such pattern, we opted to remove all such plots from the manuscript (in Fig 4 and Fig S9). Further, since center log ratio (CLR) transformation has been suggested to overcome unequal sampling depths in abundance-based RDA plots, we repeated the RDA analysis after CLR-transforming the data. The new

figures are now presented in the revised manuscript (Figure S8a and S11).

Fig. S7: which PCoA was used? (there are 3 in Fig. 4) In addition, a control is missing (euclidian distance from other centroids) to show its not an artifact of the plot itself.

Thank you. Figure S7 (now Figure S9A) was reworked with only data from the PCoA constructed with weighted Unifrac that is now showing in Figure 4. Boxplots of the distribution of Euclidean distances to group centroids are now in pink, while boxplots of the distribution of Euclidean distances to all other centroids (non-group) are in cyan.

Fig. S8: Which top 25%? abundance across all animals? In average? Needs to be clarified.

Thank you, this is now clarified in the legend to the DPCoA supplementary figure (Figure S10) and expanded in the supplementary methods.

Table S6: the table is missing.

Supplementary table 6 was originally provided as a separate excel file, and was not part of the supplementary document. This might have caused the confusion. We now have added the table to the supplementary document itself.

Again, we sincerely thank the reviewers for their excellent scientific and editorial suggestions and thorough reading of the manuscript. Such often-unappreciated efforts go a long way towards improving the readability and overall quality of published manuscripts.

We hope the implemented changes renders the manuscript suitable for publication in Nature Communications.

Yours truly,

Mostafa Elshahed

Reviewer comments, second round

Reviewer #1 (Remarks to the Author):

Thanks to the authors for their efforts in revising the manuscript. For one, I like the new panel in Figure 1 and think it will go a long way toward helping readers understand characteristics of the mammalian gut pertinent to the study.

The authors have sufficiently addressed my concerns and I would now advocate for this manuscript's publication in Nature Communications.

Reviewer #2 (Remarks to the Author):

Thanks a lot for answering my questions about the previously submitted version! You have well addressed my concerns.

Reviewer #3 (Remarks to the Author):

The authors have taken the suggestions of the three reviewers very seriously and satisfactorily answered them. The manuscript is now suitable for publication in Nature Communications.